# Modeling Heterogeneous Hierarchies with Relation-specific Hyperbolic Cones

**Yushi Bai**[*]
bys18@mails.tsinghua.edu.cn
Tsinghua University

**Rex Ying**[*]
rexying@stanford.edu
Stanford University

**Hongyu Ren**
hyren@cs.stanford.edu
Stanford University

**Jure Leskovec**
jure@cs.stanford.edu
Stanford University

## Abstract

Hierarchical relations are prevalent and indispensable for organizing human knowledge captured by a knowledge graph (KG). The key property of hierarchical relations is that they induce a partial ordering over the entities, which needs to be modeled in order to allow for hierarchical reasoning. However, current KG embeddings can model only a single global hierarchy (single global partial ordering) and fail to model multiple heterogeneous hierarchies that exist in a single KG. Here we present ConE (Cone Embedding), a KG embedding model that is able to simultaneously model multiple hierarchical as well as non-hierarchical relations in a knowledge graph. ConE embeds entities into hyperbolic cones and models relations as transformations between the cones. In particular, ConE uses cone containment constraints in different subspaces of the hyperbolic embedding space to capture multiple heterogeneous hierarchies. Experiments on standard knowledge graph benchmarks show that ConE obtains state-of-the-art performance on hierarchical reasoning tasks as well as knowledge graph completion task on hierarchical graphs. In particular, our approach yields new state-of-the-art Hits@1 of 45.3% on WN18RR and 16.1% on DDB14 (0.231 MRR). As for hierarchical reasoning task, our approach outperforms previous best results by an average of 20% across three hierarchical datasets.

## 1 Introduction

Knowledge graph (KG) is a prevalent data structure that stores factual knowledge in the form of triplets, which connect two entities (nodes) with a relation (edge) [1]. Knowledge graphs play an important role in many scientific and machine learning applications, including question answering [2], information retrieval [3] and discovery in biomedicine [4]. Knowledge graph completion is the problem of predicting missing relations in the graph, and is crucial in many real-world applications. Knowledge graph embedding (KGE) models [5, 6, 7] approach the task by embedding entities and relations into low-dimensional vector space and then use the embeddings to learn a function that given a head entity $h$ and a relation $r$ predicts the tail entity $t$.

Hierarchical information is ubiquitous in real-world KGs, such as WordNet [8] or Gene Ontology [9], since much human knowledge is organized hierarchically. The relations in these KGs can be separated into non-hierarchical relations (e.g., *likes*, *friendOf*) and hierarchical relations (e.g., *isA*,

---

[*]Equal contribution

35th Conference on Neural Information Processing Systems (NeurIPS 2021).

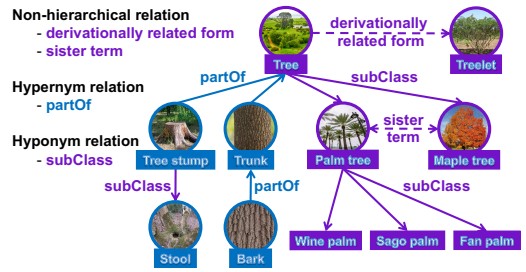

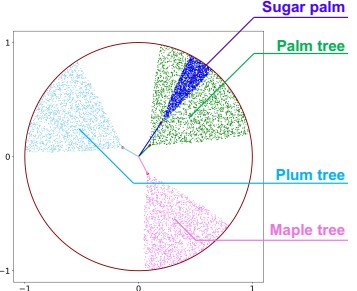

(a) Multiple heterogeneous hierarchies in knowledge graph.

(b) Hyperbolic entailment cones in 2D hyperbolic plane for $K = 0.1$.

Figure 1: (a) There are three categories of relations: non-hierarchical relation (*sister term*), hypernym (*partOf*) and hyponym relation (*subClass*). Relations induce multiple independent hierarchies. (b) ConE uses $d$ 2D hyperbolic entailment cones to model an entity. Entities *PalmTree* and *SugarPalm* are connected by a hyponym relation *subClass* and therefore the cone of *PalmTree* contains the cone of *SugarPalm*.

*partOf* ), where non-hierarchical relations capture interactions between the entities at the same level while hierarchical relations induce a tree-like partial ordering structure of entities.

Recent works propose the use of a variety of embedding geometries such as hyperbolic embeddings, box embeddings, and cone embeddings [10, 11, 12] to model partial ordering property of hierarchical relations, but two important challenges remain: **(1)** Existing works that consider hierarchical relations [13] do not take into account existing non-hierarchical relations [14]. **(2)** These methods can only be applied to graphs with a single hierarchical relation type, and are thus not suitable to real-world knowledge graphs that simultaneously encode multiple hierarchies using many different relations. For example, in Figure 1, *subClass* and *partOf* each define a unique hierarchy over the same set of entities. However, existing models treat all relations in a KG as part of one single hierarchy, limiting the ability to reason with different types of heterogeneous hierarchical relations. While there are methods for reasoning over KGs that use hyperbolic space (MuRP [15], RotH [16]), which is suitable for modeling tree-like graphs, the choice of relational transformations used in these works (rotation) prevents them from faithfully capturing all the properties of hierarchical relations. For example, they cannot model transitivity of hierarchical relations: if there exist triplets $(h_1, r, h_2)$ and $(h_2, r, h_3)$, then $(h_1, r, h_3)$ exists, *i.e.* $h_1$ and $h_3$ are also related by relation $r$.

Here we propose a novel hyperbolic knowledge graph embedding model ConE. ConE is motivated by the transitivity of nested angular cones [12] that naturally model the partial ordering defined by hierarchical relations. Our proposed approach embeds entities into the product space of hyperbolic planes, where the coordinate in each hyperbolic plane corresponds to a 2D hyperbolic cone. To address challenge **(1)**, we model non-hierarchical relations as hyperbolic cone rotations from head entity to tail entity, while we model hierarchical relations as a restricted rotation which guarantees cone containment (Figure 1(b)). To address challenge **(2)**, we assign distinct embedding subspaces corresponding to product spaces of a different set of hyperbolic planes for each hierarchical relation, to enforce cone containment constraints. By doing so, multiple heterogeneous hierarchies are preserved simultaneously in unique subspaces, allowing ConE to perform multiple hierarchical reasoning tasks accurately.

We evaluate the performance of ConE on the KG completion task and hierarchical reasoning task. A single trained ConE model can achieve remarkable performance on both tasks simultaneously. On KG completion task, ConE achieves new state-of-the-art results on two benchmark knowledge graph datasets including WN18RR [5, 17], DDB14 [18] (outperforming by **0.9%** and **4.5%** on Hits@1 metric). We also develop a novel biological knowledge graph GO21 from biomedical domain and show that ConE successfully models multiple hierarchies induced by different biological processes. We also evaluate our model against previous hierarchical modeling approaches on ancestor-descendant prediction task. Results show that ConE significantly outperforms baseline models (by **20%** on average when missing links are included), suggesting that it effectively models multiple

heterogeneous hierarchies. Moreover, ConE performs well on the lowest common ancestor (LCA) prediction task, improving over previous methods by **100%** in Hits@3 metric.

## 2 Related Work

**Hierarchical reasoning**. The most related line of work is learning structured embeddings to perform hierarchical reasoning on graphs and ontologies: order embedding, probabilistic order embedding, box embedding, Gumbel-box embedding and hyperbolic embedding [10, 11, 12, 19, 20, 21, 22]. These embedding-based methods map entities to various geometric representations that can capture the transitivity and entailment of hierarchical relations. These methods aim to perform hierarchical reasoning (transitive closure completion), such as predicting if an entity is an ancestor of another entity. However, the limitation of the above works is that they can only model a single hierarchical relation, and it remains unexplored how to extend them to multiple hierarchical relations in heterogeneous knowledge graphs. Recently, [23] builds upon the box embedding and further models joint (two) hierarchies using two boxes as entity embeddings. However, the method is not scalable since the model needs to learn a quadratic number of transformation functions between all pairs of hierarchical relations. Furthermore, the missing part is that these methods do not leverage non-hierarchical relations to further improve the hierarchy modeling. For example in Figure 1(a), with the *sisterTerm(PalmTree, MapleTree)* and *subClass(PalmTree, Tree)*, we may infer *subClass(MapleTree, Tree)*. In contrast to prior methods, ConE is able to achieve exactly this type of reasoning as it can simultaneously model multiple hierarchical as well as non-hierarchical relations.

**Knowledge graph embedding**. Various embedding methods have been proposed to model entities and relations in heterogeneous knowledge graphs. Prominent examples include TransE [5], DistMult [24], ComplEx [25], RotatE [7] and TuckER [14]. These methods often require high embedding dimensionality to model all the triples. Recently KG embeddings based on hyperbolic space have shown success in modeling hierarchical knowledge graphs. MuRP [15] learns relation-specific parameters in the Poincaré ball model. RotH [16] uses rotation and reflection transformation in $n$-dimensional Poincaré space to model relational patterns, and achieves state-of-the-art for the KG completion task, especially under low-dimensionality. However, transformations used in MuRP and RotH cannot capture transitive relations which hierarchical relations naturally are.

To the best of our knowledge, ConE is the first model that can faithfully model multiple hierarchical as well as non-hierarchical relations in a single embedding framework.

## 3 ConE Model Framework

### 3.1 Preliminaries

**Knowledge graphs and knowledge graph embeddings**. We denote the entity set and the relation set in knowledge graph as $\mathcal{E}$ and $\mathcal{R}$ respectively. Each edge in the graph is represented by a triplet $(h, r, t)$, connecting the head entity $h \in \mathcal{E}$ and the tail entity $t \in \mathcal{E}$ with relation $r \in \mathcal{R}$. In KG embedding models, entities and relations are mapped to vectors: $\mathcal{E} \to \mathbb{R}^{d_{\mathcal{E}}}, \mathcal{R} \to \mathbb{R}^{d_{\mathcal{R}}}$. Here $d_{\mathcal{E}}, d_{\mathcal{R}}$ refer to the dimensionality of entity and relation embeddings, respectively. Specifically, the mapping is learnt via optimizing a defined scoring function $\mathbb{R}^{d_{\mathcal{E}}} \times \mathbb{R}^{d_{\mathcal{R}}} \times \mathbb{R}^{d_{\mathcal{E}}} \to \mathbb{R}$ measuring the likelihood of triplets [16], while maximizing such likelihood only for true triplets.

**Hierarchies in knowledge graphs**. Many real-world knowledge graphs contain hierarchical relations [10, 11, 26]. Such hierarchical structure is characterized by very few top-level nodes corresponding to general and abstract concepts and a vast number of bottom-level nodes corresponding to concrete instances or components of the concept. Examples of hierarchical relations include *isA*, *partOf*. Note that there may exist multiple (heterogeneous) hierarchical relations in the same graph, which induce several different potentially incompatible hierarchies (i.e., partial orderings) over the same set of entities (Figure 1(a)). In contrast to prior work, our approach is able to model many simultaneous hierarchies over the same set of entities.

**Hyperbolic embeddings**. Hyperbolic embeddings can naturally capture hierarchical structures. Hyperbolic geometry is a non-Euclidean geometry with a constant negative curvature, where curvature measures how a geometric manifold deviates from Euclidean space. In this work, we use Poincaré ball model with constant curvature $c = -1$ as the hyperbolic space for entity embeddings [10].

We also investigate on more flexible curvatures, see Appendix B, results show that our model is robust enough with constant curvature $c = -1$. In particular, we denote $d$-dimensional Poincaré ball centered at origin as $\mathcal{B}^d = \{\mathbf{x} \in \mathbb{R}^d : \|\mathbf{x}\| < 1\}$, where $\|\cdot\|$ is the Euclidean norm. The Poincaré ball model of hyperbolic space is equipped with Riemannian metric:

$$g^{\mathcal{B}} = (\frac{2}{1 - \|\mathbf{x}\|^2})^2 g^E \tag{1}$$

where $g^E$ denotes the Euclidean metric, i.e., $g^E = \mathbf{I}_d$. The mobius addition $\oplus$ [27] defined on Poincaré ball model with $-1$ curvature is given by:

$$\mathbf{x} \oplus \mathbf{y} = \frac{(1 + 2\langle \mathbf{x}, \mathbf{y} \rangle + \|\mathbf{y}\|^2)\mathbf{x} + (1 - \|\mathbf{x}\|^2)\mathbf{y}}{1 + 2\langle \mathbf{x}, \mathbf{y} \rangle + \|\mathbf{x}\|^2 \|\mathbf{y}\|^2} \tag{2}$$

For each point $\mathbf{x} \in \mathcal{B}^d$, the tangent space $\mathcal{T}_{\mathbf{x}}\mathcal{B}$ is the Euclidean vector space containing all tangent vectors at $\mathbf{x}$. One can map vectors in $\mathcal{T}_{\mathbf{x}}\mathcal{B}$ to vectors in $\mathcal{B}^d$ through exponential map $\exp_{\mathbf{x}}(\cdot) : \mathcal{T}_{\mathbf{x}}\mathcal{B} \to \mathcal{B}^d$ as follows:

$$\exp_{\mathbf{x}}(\mathbf{u}) = \mathbf{x} \oplus \tanh(\frac{\|\mathbf{u}\|}{1 - \|\mathbf{x}\|})\frac{\mathbf{u}}{\|\mathbf{u}\|} \tag{3}$$

Conversely, the logarithmic map $\log_{\mathbf{x}}(\cdot) : \mathcal{B}^d \to \mathcal{T}_{\mathbf{x}}\mathcal{B}$ maps vectors in $\mathcal{B}^d$ back to vectors in $\mathcal{T}_{\mathbf{x}}\mathcal{B}$, in particular:

$$\log_{\mathbf{x}}(\mathbf{u}) = (1 - \|\mathbf{x}\|) \cdot \tanh^{-1}(\|-\mathbf{x} \oplus \mathbf{v}\|)\frac{-\mathbf{x} \oplus \mathbf{v}}{\|-\mathbf{x} \oplus \mathbf{v}\|} \tag{4}$$

Also, the hyperbolic distance between $\mathbf{x}, \mathbf{y} \in \mathcal{B}^d$ is:

$$d_{\mathcal{B}}(\mathbf{x}, \mathbf{y}) = 2\tanh^{-1}(\|-\mathbf{x} \oplus \mathbf{y}\|) \tag{5}$$

A key property of hyperbolic space is that the amount of space covered by a ball of radius $r$ in hyperbolic space increases exponentially with respect to $r$, rather than polynomially as in Euclidean space. This property contributes to the fact that hyperbolic space can naturally model hierarchical tree-like structure.

**Hyperbolic entailment cones**. Each hierarchical relation induces a partial ordering over the entities. To capture a given partial ordering, we use the hyperbolic entailment cones [12]. Figure 1(b) gives an example of 2D hyperbolic cones.

Let $\mathcal{C}_{\mathbf{x}}$ denotes the cone at apex $\mathbf{x}$. The goal is to model partial order by containment relationship between cones, in particular, the entailment cones satisfy transitivity:

$$\forall \mathbf{x}, \mathbf{y} \in \mathcal{B}^d \backslash \{\mathbf{0}\} : \ \mathbf{y} \in \mathcal{C}_{\mathbf{x}} \Rightarrow \mathcal{C}_{\mathbf{y}} \subseteq \mathcal{C}_{\mathbf{x}} \tag{6}$$

Also, for $\mathbf{x}, \mathbf{y} \in \mathcal{B}^d$, we define the angle of $\mathbf{y}$ at $\mathbf{x}$ to be the angle between the half-lines $\overrightarrow{\mathbf{o}\mathbf{x}}$ and $\overrightarrow{\mathbf{x}\mathbf{y}}$ and denote it as $\angle_{\mathbf{x}}\mathbf{y}$. It can be expressed as:

$$\angle_{\mathbf{x}}\mathbf{y} = \cos^{-1}(\frac{\langle \mathbf{x}, \mathbf{y} \rangle(1 + \|\mathbf{x}\|^2) - \|\mathbf{x}\|^2(1 + \|\mathbf{y}\|^2)}{\|\mathbf{x}\| \|\mathbf{x} - \mathbf{y}\| \sqrt{1 + \|\mathbf{x}\|^2 \|\mathbf{y}\|^2 - 2\langle \mathbf{x}, \mathbf{y} \rangle}}) \tag{7}$$

To satisfy transitivity of nested angular cones and symmetric conditions [12], we have the following expression of Poincaré entailment cone at apex $\mathbf{x} \in \mathcal{B}^d$:

$$\mathcal{C}_{\mathbf{x}} = \{\mathbf{y} \in \mathcal{B}^d | \angle_{\mathbf{x}}\mathbf{y} \le \sin^{-1}(K\frac{1 - \|\mathbf{x}\|^2}{\|\mathbf{x}\|})\} \tag{8}$$

where $K \in \mathbb{R}$ is a hyperparameter (we take $K = 0.1$). This implies that the half aperture $\phi_{\mathbf{x}}$ of cone $\mathcal{C}_{\mathbf{x}}$ is as follows:

$$\phi_{\mathbf{x}} = \sin^{-1}(K\frac{1 - \|\mathbf{x}\|^2}{\|\mathbf{x}\|}) \tag{9}$$

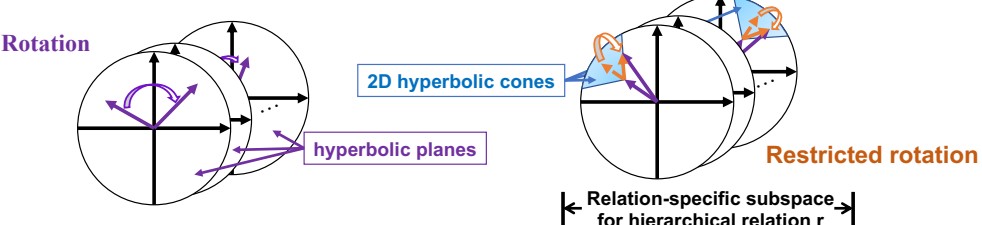

Figure 2: ConE model overview: Embedding space is the product space of $d$ hyperbolic planes and ConE learns a different transformation in each hyperbolic plane. ConE uses **restricted rotation** in an assigned relation-specific subspace to model each hierarchical relation $r$ and enforces cone containment constraint in the subspace so that partial ordering of cones is preserved in the subspace. For hyperbolic planes not in the subspace, we use a general **rotation** to model $r$. How to choose a relation-specific subspace for each hierarchical relation is essential and further explained in Sec. 3.3.

## 3.2    ConE Embedding Space and Transformations

We first introduce the embedding space that ConE operates in, and the transformations used to model hierarchical as well as non-hierarchical relations.

For ease of discussion let's assume that the relation type is given *a priori*. In fact, knowledge about hierarchical relations (i.e., transitive closure) is explicitly available in the definition of the relation in KGs such as ConceptNet [28], WordNet [8] and Gene Ontology [9]. When such information is not available, ConE can infer "hierarchicalness" of a relation by a simple criteria with slight modification to the Krackhardt scores [29], see Appendix H.

**Embedding space**. The embedding space of ConE, $\mathcal{S}$, is a product space of $d$ hyperbolic planes [30], resulting in a total embedding dimension of $2d$. $\mathcal{S}$ can be denoted as $\mathcal{S} = \mathcal{B}^2 \times \mathcal{B}^2 \times \cdots \times \mathcal{B}^2$. Note that this space is different from RotH embedding space [16], which is a single $2d$-dimensional hyperbolic space. ConE's embedding space is critical in modeling ancestor-descendant relationships for heterogeneous KGs, since it is more natural when allocating its subspaces (product space of multiple hyperbolic planes) to heterogeneous hierarchical relations.

We denote the embedding of entity $h \in \mathcal{E}$ as $\mathbf{h} = (\mathbf{h}_1, \mathbf{h}_2, \cdots, \mathbf{h}_d)$ where $\mathbf{h}_i \in \mathcal{B}^2$ is the apex of the $i$-th 2D hyperbolic cone. We model relation $r$ as a cone transformation on each hyperbolic plane from head entity cone to tail entity cone. Let $\mathbf{r} = (\mathbf{r}_1, \mathbf{r}_2, \cdots, \mathbf{r}_d)$ be the representation of relation $r$. We use $\mathbf{r}_i = (s_i, \theta_i)$ to parameterize transformation for the $i$-th hyperbolic plane as shown in Figure 2. $s_i > 0$ is the scaling factor indicating how far to go in radial direction and $(\theta_i \cdot \phi_{\mathbf{h}_i}/\pi)$ is the rotation angle restricted by half aperture $\phi_{\mathbf{h}_i}$ ($\theta_i \in [-\pi, \pi)$). To perform hierarchical tasks such as ancestor-descendant prediction, ConE uses nested cones in each hyperbolic plane to model the partial ordering property of hierarchical relations, by the cone containment constraint in Def. 1.

**Definition 1.** *Cone containment constraint. If entity $h$ is an ancestor of $t$, then the cone embedding of $t$ has to reside in that of the entity $h$, i.e., $\mathcal{C}_{\mathbf{t}_i} \subseteq \mathcal{C}_{\mathbf{h}_i}, \forall i \in \{1, ...d\}$.*

The cone containment constraint can be enforced in any of the hyperbolic plane components in $\mathcal{S}$. Next we introduce ConE's transformations for characterizing hierarchical and non-hierarchical patterns of relation $r$ in triple $(h, r, t)$. Note that we utilize both transformations to model hierarchical relations $r$ to capture non-hierarchical properties, i.e., symmetry, composition, etc, as well as hierarchical properties, i.e., partial ordering. We do this by performing different transformations in different subspaces of $\mathcal{S}$, as discussed in detail in Sec. 3.3.

**Transformation for modeling non-hierarchical properties**. Rotation is an expressive transformation to capture relation between entities [7]. Analogous to RotatE, we adopt **rotation transformation** $f_1$ to model non-hierarchical properties (Figure 3(a)). For rotation in the $i$-th hyperbolic plane,

$$f_1(\mathbf{h}_i, \mathbf{r}_i) = \exp_{\mathbf{o}}(\mathbf{G}(\theta_i) \log_{\mathbf{o}}(\mathbf{h}_i)) \tag{10}$$

where $\mathbf{G}(\theta_i)$ is the Givens rotation matrix:

$$\mathbf{G}(\theta_i) = \begin{bmatrix} \cos(\theta_i) & -\sin(\theta_i) \\ \sin(\theta_i) & \cos(\theta_i) \end{bmatrix} \tag{11}$$

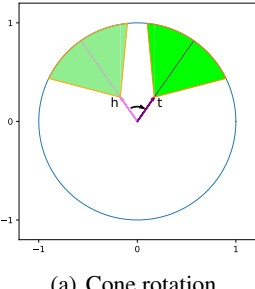

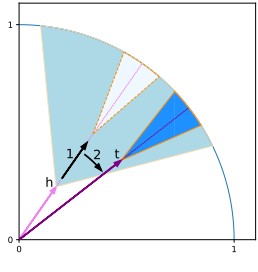

|  (a) Cone rotation | (b) Restricted cone rotation |

Figure 3: Transformations in ConE in Poincaré ball: (a) Cone rotation from $h$ to $t$ used for non-hierarchical relations; (b) Restricted rotation from the cone of parent $h$ to the cone of child $t$ used for hierarchical relations, where "1" corresponds to scaling and "2" to rotation $(s_i, \theta_i)$ in Eq. 12.

We also show that the rotation transformation in Eq. 10 is expressive: It can model relation patterns including symmetry, anti-symmetry, inversion, and composition (Appendix A.1).

**Transformation for modeling hierarchical properties**. However, $f_1$ cannot be directly applied to model hierarchical relations, because rotation does not obey transitive property: rotation by $\theta_i$ twice will result in a rotation of $2\theta_i$, instead of $\theta_i$. Hence it cannot guarantee $(h_1, r, h_3)$ when $(h_1, r, h_2)$ and $(h_2, r, h_3)$ are true. We use **restricted rotation transformation** $f_2$ to model hierarchical relations. We impose cone containment constraint to preserve partial ordering of cones after the transformation. Without loss of generality we assume relation $r$ is a hyponym type relation, the restricted rotation from $h$ to $t$ in $i$-th hyperbolic plane is as follows (we perform restricted rotation from $t$ to $h$ if $r$ is a hypernym relation):

$$f_2(\mathbf{h}_i, \mathbf{r}_i) = \exp_{\mathbf{h}_i}(s_i \cdot \mathbf{G}(\theta_i \frac{\phi_{\mathbf{h_i}}}{\pi})\overline{\mathbf{h}}_i), \ \mathbf{r}_i = (s_i, \theta_i) \tag{12}$$

where $\phi_{\mathbf{h_i}}$ is the half aperture of cone $\mathbf{h}_i$. $\overline{\mathbf{h}}_i$ is the unit vector of $\mathbf{h}_i$ in the tangent space of $\mathbf{h}_i$:

$$\overline{\mathbf{h}}_i = \widehat{\mathbf{h}}_i / ||\widehat{\mathbf{h}}_i||, \ \widehat{\mathbf{h}}_i = \log_{\mathbf{h}_i}(\frac{1 + ||\mathbf{h}_i||}{2||\mathbf{h}_i||}\mathbf{h}_i) \tag{13}$$

Figure 3(b) illustrates the two-step transformation described in Eq. 12, namely the scaling step and the rotation step.

### 3.3 ConE Model of Heterogeneous Hierarchies

In the previous section, we explained how we enforce cone containment constraint for hierarchical relations, however two challenges remain when simultaneously modeling multiple heterogeneous hierarchies: **(1) Partial ordering:** Suppose that there is a hyponym relation between entities $h_1$ and $h_2$, and a *different* hyponym relation between entities $h_2$ and $h_3$. Then a naïve model would enforce that the cone of $h_1$ contains the cone of $h_2$ which contains the cone of $h_3$, implying that a hyponym relation exists between $h_1$ and $h_3$, which is not correct. **(2) Expressive power:** Cone containment constraint, while ensuring hierarchical structure by geometric entailment, limits the set of possible rotation transformations and thus limits the model's expressive power.

To address these challenges we proceed as follows. Instead of enforcing cone containment constraint in the entire embedding space, ConE proposes a novel technique to assign unique subspace for each hierarchical relation, *i.e.* we enforce cone containment constraint only in a subset of $d$ hyperbolic planes. Next we further elaborate on this idea.

In particular, for a hierarchical relation $r$, we assign a corresponding subspace of $\mathcal{S}$, which is a product space of a subset of hyperbolic planes. Then, we use restricted rotation in the subspace and rotation in the complement space. We train ConE to enforce cone containment constraint in the relation-specific subspace. The subspace can be represented by a $d$-dimensional mask $\mathbf{m}, \mathbf{m}_i \in \{0, 1\}$, and $\mathbf{m}_i = 1$ indicates that cone containment is enforced in the $i$-th hyperbolic plane. We then extend such notation to all relations where $\mathbf{m} = \mathbf{0}$ for non-hierarchical relations.

| Dataset | #entities | #relations | #training | #validation | #test | Examples of hierarchical relations |
|---------|-----------|------------|-----------|-------------|-------|-----------------------------------|
| WN18RR | 40,943 | 11 | 86,385 | 3,034 | 3,134 | *hypernym, has part* |
| DDB14 | 9,203 | 14 | 38,233 | 4,000 | 4,000 | *subtype of, subset of* |
| GO21 | 89,127 | 21 | 796,136 | 5,000 | 5,000 | *part of, is a* |
| FB15k-237 | 14,541 | 237 | 272,115 | 17,535 | 20,466 | *location/contains, /music/genre/parent_genre* |

Table 1: Datasets statistics. Note that FB15k-237 has very few such hierarchical relations.

Our design of leveraging both transformations to model hierarchical relations is crucial in that they capture different aspects of the relation. The use of restricted rotation along with cone containment constraint serves to preserve partial ordering of a hierarchical relation in its relation-specific subspace. But restricted rotation alone is insufficient: hierarchical relations also possess other properties such as composition and symmetry that cannot be modeled by restricted rotation. Hence we augment with the rotation transformation to capture these properties, allowing composition of different hierarchical and non-hierarchical relations through rotations in the complement space. We further provide theoretical and empirical results in Appendix A to support that both transformations are of great significance to the expressiveness of our model.

Putting it all together gives us the following distance scoring function (we use $(v_i)_{i \in \{1, \cdots, d\}}$ in the following to denote a $d$-dimensional vector $\mathbf{v}$):

$$\psi(h, r, t) = -\frac{1}{d}[\mathbf{m} \cdot (d_{\mathcal{B}}(f_2(\mathbf{h}_i, \mathbf{r}_i), \mathbf{t}_i))_{i \in \{1, \cdots, d\}}$$
$$+(\mathbf{1} - \mathbf{m}) \cdot (d_{\mathcal{B}}(f_1(\mathbf{h}_i, \mathbf{r}_i), \mathbf{t}_i))_{i \in \{1, \cdots, d\}}] + b_h + b_t \tag{14}$$

where the first term corresponds to the restricted rotation in relation-specific subspace, and the second term corresponds to the rotation in complementary space. A high score indicates that cone of entity $h$ after relation-specific transformation $r$ is close to the cone of entity $t$ in terms of hyperbolic distance $d_{\mathcal{B}}$. Note that $b_h, b_t$ are the learnt radius parameters of $h, t$ which can be interpreted as margins [15].

**Subspace allocation**. We assign equal dimensional subspaces for all hierarchical relations. We discuss and compare several strategies in assigning subspaces for hierarchical relations in Appendix B, including whether to use overlapping subspaces or orthogonal subspaces for different hierarchical relations, as well as the choice of dimensionality of subspaces. Overlapping subspaces (Appendix B) allow the model to perform well and enable it to scale to knowledge graphs with a large number of relations, since there are exponentially many possible overlapping subspaces that can potentially correspond to different hierarchical relations.

### 3.4 ConE Loss Function

We use a loss function composed of two parts. The first part of the loss function aims to ensure that for a given head entity $h$ and relation $r$ the distance to the true tail entity $t$ is smaller than to the negative tail entity $t'$:

$$\mathcal{L}_d(h, r, t) = -\log \sigma(\psi(h, r, t)) - \sum_{t' \in \mathcal{T}} \frac{1}{|\mathcal{T}|} \log \sigma(-\psi(h, r, t')) \tag{15}$$

where $(h, r, t)$ denotes a positive training example/triplet, and we generate negative samples $(h, r, t')$ by substituting the tail with a random entity in $\mathcal{T} \subseteq \mathcal{E}$, a random set of entities in KG excluding $t$.

However, the distance loss $\mathcal{L}_d$ does not guarantee embeddings satisfying the cone containment constraint, since the distance between transformed head embedding and tail embedding can still be non-zero after training. Hence we additionally introduce the **angle loss** (without loss of generality let $r$ be a hyponym relation):

$$\mathcal{L}_a(h, r, t) = \mathbf{m} \cdot (\max(0, \angle_{\mathbf{h}_i} \mathbf{t}_i - \phi(\mathbf{h}_i)))_{i \in \{1, \cdots, d\}} \tag{16}$$

which directly encourages cone of $h$ to contain cone of $t$ in relation-specific subspaces, by constraining the angle between the cones. The final loss is then a weighted sum of the distance loss and the angle loss, where weight $w$ is a hyperparameter (We investigate the choice of $w$ in Appendix B):

$$\mathcal{L} = \mathcal{L}_d + w \cdot \mathcal{L}_a \tag{17}$$

| | WN18RR | | | DDB14 | | | GO21 | | |
|---|---|---|---|---|---|---|---|---|---|
| | Fraction of inferred descendant pairs among all true descendant pairs in the test set | | | | | | | | |
| Model | 0% | 50% | 100% | 0% | 50% | 100% | 0% | 50% | 100% |
| Order [19] | .889 | .739 | .498 | .731 | .633 | .513 | .642 | .592 | .534 |
| Poincaré [10] | .810 | .685 | .508 | .976 | .832 | .571 | .525 | .519 | .516 |
| HypCone [12] | .799 | .677 | .504 | .973 | .823 | .594 | .554 | .539 | .519 |
| RotatE [7] | .601 | .593 | .582 | .615 | .590 | .565 | .546 | .534 | .526 |
| RotH [16] | .601 | .608 | .611 | .609 | .596 | .578 | .596 | .583 | .564 |
| ConE | **.895** | **.801** | **.679** | **.981** | **.909** | **.818** | **.789** | **.744** | **.693** |
| Improvement (%) | +1.9% | +9.6% | +11.1% | +0.5% | +10.3% | +38.4% | +22.9% | +25.7% | +22.9% |

Table 2: Ancestor-descendant prediction results in mAP (mean average precision). Best score in **bold** and second best underlined. We create different test sets that get harder as they contain more and more test cases (0%, 50%, 100%) of inferred descendant pairs.

## 4 Experiments

Given a KG containing many hierarchical and non-hierarchical relations, our experiments evaluate: **(A)** Performance of ConE on hierarchical reasoning task of predicting if entity $h_1$ is an ancestor of entity $h_2$. **(B)** Performance of ConE on generic KG completion tasks.

**Datasets**. We use four knowledge graph benchmarks (Table 1): WordNet lexical knowledge graph (WN18RR [5, 17]), drug knowledge graph (DDB14 [18]), and a KG capturing common knowledge (FB15k-237 [31]). Furthermore, we also curated a new biomedical knowledge graph GO21, which models genes and the hierarchy of biological processes they participate in.

**Model training**. During training, we use Adam [32] as the optimizer and search hyperparameters including batch size, embedding dimension, learning rate, angle loss weight and dimension of subspace for each hierarchical relation. (Training details and standard deviations in Appendix G).[2]

We use a **single** trained model (without fine-tuning) for all evaluation tasks: On ancestor-descendant relationship prediction, our scoring function for a pair $(h, t)$ with hierarchical relation $r$ is the angle loss in Eq. 16 where a lower score means $h$ is more likely to be an ancestor of $t$. For KG completion task we use the scoring function $\psi(h, r, t)$ in Eq. 14 to rank the triples.

### 4.1 Hierarchical Reasoning: Ancestor-descendant Prediction

Next we define ancestor-descendant relationship prediction task to test model's ability on hierarchical reasoning. Given two entities, the goal makes a binary prediction if they have ancestor-descendant relationship:

**Definition 2.** *Ancestor-descendant relationship. Entity pair $(h_1, h_2)$ is considered to have ancestor-descendant relationship if: there exists a path from $h_1$ to $h_2$ that only contains one type of hyponym relation, or a path from $h_2$ to $h_1$ that only contains one type of hypernym relation.*

Our evaluation setting is a generalization of the transitive closure prediction [19, 10, 12] which is defined only over a single hierarchy, but our knowledge graphs contain multiple hierarchies (hierarchical relations). More precisely: **(1)** When heterogeneous hierarchies coexist in the graph, we compute the transitive closure induced by each hierarchical relation separately. The test set for each hierarchical relation is a random collection sampled from all transitive closures of that relation. **(2)** To increase the difficulty of the prediction task, our evaluation also considers **inferred descendant pairs**, which are only possible to be inferred when simultaneously considering hierarchical and non-hierarchical relations in KG, due to missing links in KG. We call a descendant pair $(u, v)$ an inferred descendant pair if their ancestor-descendant relationship can be inferred from the whole graph but *not* from the training set. For instance, *(Tree,WinePalm)* would be an *inferred descendant pair* if the *subClass* relation between *Tree* and *PalmTree* is missing in training set. We construct the inferred descendant pairs by taking the transitive closures of the entire graph, and exclude the transitive closures of relations in the training set. In our experiments, we consider three test settings: 0%, 50%, 100%, corresponding to the fraction of inferred descendant pairs among all true descendant pairs in the test set, and the setting with a higher fraction is harder.

---

[2]The code of our paper is available at `http://snap.stanford.edu/cone`.

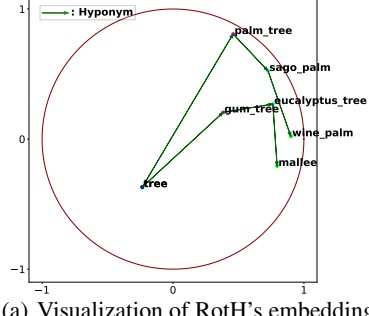

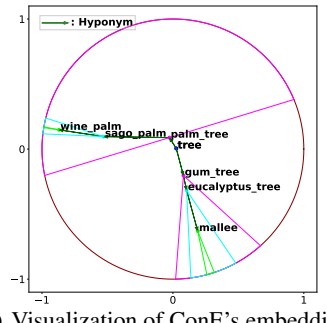

| (a) Visualization of RotH's embedding | (b) Visualization of ConE's embedding |

Figure 4: The embeddings of RotH and ConE, trained on WN18RR, projecting to one hyperbolic plane. We show the embedding of a family of trees, and the arrows point from higher level entities to lower level entities, representing the hierarchical relation "*Hyponym*". Different levels of entities and their corresponding cones in ConE model (Figure 4(b)) are marked with different colors. In ConE model, the embeddings of high-level entities (e.g., tree, palm tree) are close to the center of the hyperbolic plane, while embeddings of their descendant entities (e.g., wine palm, mallee) fall in their hyperbolic cones.

On each dataset, we extract 50k ancestor-descendant pairs. For each pair, we randomly replace the true descendant with a random entity in the graph, resulting in a total of 100k pairs. Our way of selecting negative examples offsets the bias during learning that is prevalent in baseline models: the models tend to always give higher scores to pairs with a high-level node as ancestor, since high-level nodes usually have more descendants presented in training data. We replace the true descendant while keeping the true ancestor unchanged for the negative sample, and thus the model will not be able to "cheat" by taking advantage of the fore-mentioned bias. For each model, we then use its scoring function to rank all the pairs. We use the standard mean average precision (mAP) to evaluate the performance on this binary classification task. We further show the AUROC results in Appendix E.

**Baselines**. We compare our method with state-of-the-art methods for hierarchical reasoning, including Order embeddings [19], Poincaré embeddings [10] and Hyperbolic entailment cones [12]. Note that these methods can only handle a single hierarchical relation at a time. So each baseline trains a separate embedding for each hierarchical relation and then learns a scoring function on the embedding of the two entities. To ensure that the experiment controls the model size, we enforce that in baselines, the sum of embedding dimensions of all relations is equal to the relation embedding dimension of ConE. We also perform comprehensive hyperparameter search for all baselines (Appendix G). Although KG embedding models (RotatE [7] and RotH [16]) cannot be directly applied to this task, we adapt them to perform this task by separately training an MLP to make binary classification on ancestor-descendant pair, taking the concatenation of the two entity embeddings as input. Note that ConE outperforms these KG completion methods without even requiring additional training.

**Results**. Table 2 reports the ancestor-descendant prediction results of ConE and the baselines. We observe that the novel subspace transformation of ConE results in its superior performance in this task. Our model consistently outperforms baseline methods on all three datasets. As we expected, KG embedding models cannot perform well on this task (in the range of $0.5 \sim 0.6$ across all settings), since they do not explicitly model the partial ordering property of the hierarchical relations. In contrast, our visualization of ConE's embedding in Figure 4 suggests that ConE faithfully preserves the cone containment constraint in modeling hierarchical relations, while RotH's embedding exhibit less hierarchical structure. As a result, ConE simultaneously captures the heterogeneous relation modeling and partial ordering, combining the best of both worlds. Our improvement is more significant as the fraction of inferred descendant pairs increases. This shows that ConE not only embeds a given hierarchical structure, but can also infer missing hierarchical links by modeling other non-hierarchical relations at the same time. Thanks to the restricted rotation transformation and the use of product spaces of hyperbolic planes, ConE can faithfully model the hierarchies without requiring all transitive closures in the training set. We further perform additional studies to explore reasons for the performance of each method on ancestor-descendant prediction task in Appendix E.

**Lowest common ancestor prediction task**. Moreover, we demonstrate flexibility and power of ConE using a hierarchical analysis task: lowest common ancestor (LCA) prediction, which requires both the

| | WN18RR $\kappa = (1.00, 0.61, 0.99, 0.50)$ | | | | DDB14 $\kappa = (1.00, 0.84, 0.78, 0.18)$ | | | | GO21 $\kappa = (1.00, 0.65, 0.96, 0.22)$ | | | | FB15k-237 $\kappa = (1.00, 0.18, 0.36, 0.06)$ | | | |
|---|---|---|---|---|---|---|---|---|---|---|---|---|---|---|---|---|
| Model | MRR | H@1 | H@3 | H@10 | MRR | H@1 | H@3 | H@10 | MRR | H@1 | H@3 | H@10 | MRR | H@1 | H@3 | H@10 |
| TransE [5] | .226 | .017 | .403 | .532 | .183 | .103 | .212 | .337 | .149 | .066 | .179 | .310 | .294 | - | - | .465 |
| RotatE [7] | .476 | .428 | .429 | .571 | .225 | .154 | .245 | .362 | .203 | .123 | .234 | **.357** | .338 | .241 | .375 | .533 |
| TuckER [14] | .470 | .443 | .482 | .526 | .198 | .137 | .219 | .314 | .205 | .136 | .222 | .342 | **.358** | **.266** | **.394** | **.544** |
| HAKE [33] | .496 | .451 | .513 | .582 | .217 | .146 | .237 | .361 | .169 | .104 | .185 | .295 | .341 | .243 | .378 | .535 |
| MuRP [15] | .481 | .440 | .495 | .566 | .214 | .146 | .231 | .349 | .166 | .100 | .181 | .301 | .335 | .243 | .367 | .518 |
| RotH [16] | .495 | .449 | .514 | .586 | .223 | .152 | .245 | .357 | .151 | .079 | .171 | .289 | .344 | .246 | .380 | .535 |
| ConE | .496 | **.453** | **.515** | .579 | **.231** | **.161** | **.252** | **.364** | .211 | **.140** | **.237** | .347 | .345 | .247 | .381 | .540 |

Table 3: Knowledge graph completion results, best out of dimension $d \in \{100, 250, 500\}$. Best score in **bold** and second best underlined. $\kappa$ is a tuple denoting the 4 Krackhardt scores [29] that measure how hierarchical a graph is, higher scores mean more hierarchical. ConE achieves the best MRR and Hits@1 results in hierarchical KGs.

ability to model ancestor-descendant relationship and to distinguish the lowest ancestor. Results show that ConE can precisely predict LCA, outperforming over 100% on Hits@3 and Hits@10 metrics compared to previous methods (See detailed results and analysis in Appendix F).

## 4.2 Knowledge Graph Completion

We also experiment on knowledge graph completion task where missing links include hierarchical relations as well as non-hierarchical relations. We follow the standard evaluation setting [5].

**Baselines**. We compare ConE model to state-of-the-art models on knowledge graph completion task, including TransE [5], RotatE [7], TuckER [14] and HAKE [33], as well as MuRP [15] and RotH [16], which both operate on a hyperbolic space.

**Results**. Table 3 reports the KG completion results. Over the first three hierarchical datasets considered, ConE achieves state-of-the-art results over many recent baselines, including the recently proposed hyperbolic approaches RotH and MuRP. We also notice that the margins on Hits@1 and Hits@3 scores are much larger than Hits@10, indicating that our model provides the most accurate predictions. We further use Krackhardt scores $\kappa$ to measure how hierarchical each graph is [29]. The score consists of four metrics (*(connectedness, hierarchy, efficiency, LUBedness)*, Appendix H), where if a graph is maximally hierarchical (i.e., a tree) then its Krackhardt score is $(1, 1, 1, 1)$, and higher score on four metrics indicate a more hierarchical structure. Notice that the Krackhardt scores of FB15k-237 are approximately three times lower than those of WN18RR, DDB14 and GO21, indicating that FB15k-237 is indeed non-hierarchical. We can see that our ConE model still performs better than other hierarchical KG embedding models (RotH and MuRP) on FB15k-237 and is comparable to SOTA model (TuckER). Overall, this shows that ConE can scale to a large number of relations, and that it has competitive performance even in non-hierarchical knowledge graphs.

We further analyze the performance of ConE in low-dimensional regimes in Appendix C. Similar to previous studies, the hyperbolic-space-based ConE model performs much better than Euclidean KG embeddings in low dimensions ($d = 32$). ConE performs similar to previous hyperbolic KG embedding baselines in low dimensions, but outperforms them in high-dimensional regimes (Table 2).

**Ablation study**. We further compare the performance of our model with one that does not use cone restricted rotation for modeling hierarchical relations and one that does not use rotation for modeling hierarchical relations. Ablation results suggest that both transformations, i.e., cone restricted rotation and rotation, are critical in predicting missing hierarchical relations (Appendix A.2). In particular, our ablation results on each individual hierarchical relation suggest that with cone restricted rotation, ConE can simultaneously model heterogeneous hierarchical relations effectively.

## 5 Conclusion

In this paper, we propose ConE, a hierarchical KG embedding method that models entities as hyperbolic cones and uses different transformations between cones to simultaneously capture hierarchical and non-hierarchical relation patterns. We apply cone containment constraint to relation-specific subspaces to capture hierarchical information in heterogeneous knowledge graphs. ConE can simultaneously perform knowledge graph completion task and hierarchical task, and achieves state-of-the-art results on both tasks across three hierarchical knowledge graph datasets.

## Acknowledgments and Disclosure of Funding

We gratefully acknowledge the support of DARPA under Nos. HR00112190039 (TAMI), N660011924033 (MCS); ARO under Nos. W911NF-16-1-0342 (MURI), W911NF-16-1-0171 (DURIP); NSF under Nos. OAC-1835598 (CINES), OAC-1934578 (HDR), CCF-1918940 (Expeditions), IIS-2030477 (RAPID), NIH under No. R56LM013365; Stanford Data Science Initiative, Wu Tsai Neurosciences Institute, Chan Zuckerberg Biohub, Amazon, JPMorgan Chase, Docomo, Hitachi, Intel, JD.com, KDDI, NVIDIA, Dell, Toshiba, Visa, and UnitedHealth Group. Hongyu Ren is supported by the Masason Foundation Fellowship and the Apple PhD Fellowship. Jure Leskovec is a Chan Zuckerberg Biohub investigator.

The content is solely the responsibility of the authors and does not necessarily represent the official views of the funding entities.

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
