# A  Theoretical and empirical evidence for ConE's design choice

Here we provide theoretical and empirical results to support that ConE's design choice makes sense, i.e., both rotation transformation and restricted transformation play a crucial role to the expressiveness of the model.

## A.1  Proof for transformations

### A.1.1  Proof for rotation transformation

We will show that the rotation transformation in Eq. 10 can model all relation patterns that can be modeled by its Euclidean counterpart RotatE [7].

Three most common relation patterns are discussed in [7], including symmetry pattern, inverse pattern and composition pattern. Let $\mathbb{T}$ denote the set of all true triples. We formally define the three relation patterns as follows.

**Definition 3.** *If a relation $r$ satisfies symmetric pattern, then*

$$\forall h, t \in \mathcal{E}, \ (h, r, t) \in \mathbb{T} \Rightarrow (t, r, h) \in \mathbb{T}$$

**Definition 4.** *If relation $r_1$ and $r_2$ satisfies inverse pattern, i.e., $r_1$ is inverse to $r_2$, we have*

$$\forall h, t \in \mathcal{E}, \ (h, r_1, t) \in \mathbb{T} \Rightarrow (t, r_2, h) \in \mathbb{T}$$

**Definition 5.** *If relation $r_1$ is composed of $r_2$ and $r_3$, then they satisfies composition pattern,*

$$\forall h, m, t \in \mathcal{E}, \ (h, r_2, m) \in \mathbb{T} \wedge (m, r_3, t) \in \mathbb{T} \Rightarrow (h, r_1, t) \in \mathbb{T}$$

**Theorem 1.** *Rotation transformation can model symmetric pattern.*

*Proof.* If $r$ is a symmetric relation, then for each triple $(h, r, t)$, its symmetric triple $(t, r, h)$ is also true. For $i \in \{1, 2, \cdots, d\}$, we have

$$\mathbf{t}_i = \exp_{\mathbf{o}}(\mathbf{G}(\theta_i) \log_{\mathbf{o}}(\mathbf{h}_i)), \ \mathbf{h}_i = \exp_{\mathbf{o}}(\mathbf{G}(\theta_i) \log_{\mathbf{o}}(\mathbf{t}_i))$$

Let $\mathbf{I}$ denote the identity matrix. By taking logarithmic map on both sides, we have

$$\log_{\mathbf{o}}(\mathbf{t}_i) = \mathbf{G}(\theta_i) \log_{\mathbf{o}}(\mathbf{h}_i), \ \log_{\mathbf{o}}(\mathbf{h}_i) = \mathbf{G}(\theta_i) \log_{\mathbf{o}}(\mathbf{t}_i) \ \Rightarrow \ \mathbf{G}^2(\theta_i) = \mathbf{I}$$

which holds true when $\theta_i = -\pi$ or $\theta_i = 0$ (still we assume $\theta_i \in [-\pi, \pi)$).

**Theorem 2.** *Rotation transformation can model inverse pattern.*

*Proof.* If $r_1$ and $r_2$ are inverse relations, then for each triple $(h, r_1, t)$, its inverse triple $(t, r_2, h)$ also holds. Let $(\theta_i)_{i \in \{1, \cdots, d\}}$ denote the rotation parameter of relation $r_1$ and $(\alpha_i)_{i \in \{1, \cdots, d\}}$ denote the rotation parameter of relation $r_2$. Similar to the proof of Theorem 1, we take logarithmic map on rotation transformation, then

$$\log_{\mathbf{o}}(\mathbf{t}_i) = \mathbf{G}(\theta_i) \log_{\mathbf{o}}(\mathbf{h}_i), \ \log_{\mathbf{o}}(\mathbf{h}_i) = \mathbf{G}(\alpha_i) \log_{\mathbf{o}}(\mathbf{t}_i) \ \Rightarrow \ \mathbf{G}(\theta_i) \mathbf{G}(\alpha_i) = \mathbf{I}$$

which holds true when $\theta_i + \alpha_i = 0$.

**Theorem 3.** *Rotation transformation can model composition pattern.*

*Proof.* If relation $r_1$ is composed of $r_2$ and $r_3$, then triple $(h, r_1, t)$ exists when $(h, r_2, m)$ and $(m, r_3, t)$ exist. Let $(\theta_i)_{i \in \{1, \cdots, d\}}$, $(\alpha_i)_{i \in \{1, \cdots, d\}}$, $(\beta_i)_{i \in \{1, \cdots, d\}}$, denote their rotation parameters correspondingly. Still we take logarithmic map on rotation transformation and it can be derived that

$$\log_{\mathbf{o}}(\mathbf{t}_i) = \mathbf{G}(\theta_i) \log_{\mathbf{o}}(\mathbf{h}_i), \ \log_{\mathbf{o}}(\mathbf{m}_i) = \mathbf{G}(\alpha_i) \log_{\mathbf{o}}(\mathbf{h}_i),$$
$$\log_{\mathbf{o}}(\mathbf{t}_i) = \mathbf{G}(\beta_i) \log_{\mathbf{o}}(\mathbf{m}_i) \ \Rightarrow \ \mathbf{G}(\theta_i) = \mathbf{G}(\alpha_i) \mathbf{G}(\beta_i)$$

which holds true when $\theta_i = \alpha_i + \beta_i$ or $\theta_i = \alpha_i + \beta_i + 2\pi$ or $\theta_i = \alpha_i + \beta_i - 2\pi$.

| | All relations | | | Hierarchical relations | | | Non-hierarchical relations | | |
|---|---|---|---|---|---|---|---|---|---|
| Model | MRR | H@1 | H@10 | MRR | H@1 | H@10 | MRR | H@1 | H@10 |
| RotC | .481 | .444 | .551 | .209 | .157 | .312 | .936 | .923 | .951 |
| ConE | **.496** | **.453** | **.579** | **.231** | **.171** | **.355** | **.939** | **.930** | **.952** |
| Improvement (%) | +3.1% | +2.0% | +5.1% | +10.5% | +8.9% | +13.8% | +0.3% | +0.7% | +0.1% |

Table 4: Results of ablation study on restricted rotation, for knowledge graph completion task on WN18RR. Results in three columns are conducted on all relations during evaluation, only hierarchical relations during evaluation and only non-hierarchical relations during evaluation.

| Relation | RotC | ConE | Improvement |
|---|---|---|---|
| *hypernym* | .175 | **.193** | +10.3% |
| *instance hypernym* | .373 | **.406** | +8.8% |
| *member meronym* | .230 | **.231** | +0.4% |
| *synset domain topic of* | .382 | **.413** | +8.1% |
| *has part* | .208 | **.213** | +2.4% |
| *member of domain usage* | .200 | **.345** | +72.5% |
| *member of domain region* | .142 | **.244** | +71.8% |

Table 5: Comparison of MRR for all hierarchical relations in WN18RR between RotC and ConE.

## A.1.2 Proof for restricted rotation transformation

**Theorem 4.** *Restricted rotation transformation always satisfies the cone containment constraint.*

*Proof.* For any triple $(h, r, t)$ where $r$ is a hierarchical relation, we will prove that cone containment constraint is satisfied after the restricted rotation from $h$ to $t$, i.e., $\mathcal{C}_{f_2(\mathbf{h}_i, \mathbf{r}_i)} \subseteq \mathcal{C}_{\mathbf{h}_i}$. By the transitivity property of entailment cone as in Eq. 6, we only need to prove $f_2(\mathbf{h}_i, \mathbf{r}_i) \in \mathcal{C}_{\mathbf{h}_i}$, which is

$$\angle_{\mathbf{h}_i} f_2(\mathbf{h}_i, \mathbf{r}_i) \leq \phi_{\mathbf{h}_i} \tag{18}$$

according to the cone expression in Eq. 8. We can calculate the angle, denoted as $\varphi$, on the left hand side of the equation in tangent space $\mathcal{T}_{\mathbf{h}_i}\mathcal{B}$ (which is equipped with Euclidean metric),

$$\begin{aligned}
\varphi &= \angle_{\mathbf{h}_i} f_2(\mathbf{h}_i, \mathbf{r}_i) \\
&= \angle(\log_{\mathbf{h}_i}(\frac{1 + ||\mathbf{h}_i||}{2||\mathbf{h}_i||}\mathbf{h}_i), \log_{\mathbf{h}_i} f_2(\mathbf{h}_i, \mathbf{r}_i)) \\
&= \angle(\overline{\mathbf{h}}_i, \mathbf{G}(\theta_i \frac{\phi_{\mathbf{h}_i}}{\pi})\overline{\mathbf{h}}_i) = |\theta_i \frac{\phi_{\mathbf{h}_i}}{\pi}|
\end{aligned} \tag{19}$$

For $\theta_i \in [-\pi, \pi)$, we have $|\theta_i \frac{\phi_{\mathbf{h}_i}}{\pi}| \leq \phi_{\mathbf{h}_i}$. Therefore Eq. 18 holds, suggesting that cone containment constraint is satisfied.

## A.2 Ablation studies on transformations in ConE

Empirically, we show that our design of transformations in ConE is effective: both restricted rotation transformation in the relation-specific subspace and the rotation transformation in the complement space are indispensable to the performance of our model on knowledge graph completion task.

### A.2.1 Ablation study on restricted rotation transformation

Restricted rotation transformation is vital in enforcing cone containment constraint, and thus it is indispensable to ConE's performance on hierarchical tasks. However, its effect on knowledge graph completion task remains unknown. We further compare the performance of ConE with one that does not use cone restricted rotation for modeling hierarchical relations, which we name as RotC. Specifically, RotC is the same as ConE, except that it applies rotation transformation to all relations, and the cone angle loss as in Eq. 16 is excluded.

**Results**. Ablation results are shown in Table 4. We can see remarkable improvement on knowledge graph completion task after applying restricted rotation transformation to hierarchical relations,

| Model | MRR | H@1 | H@3 | H@10 |
|---|---|---|---|---|
| ConE w/o rotation | .397 | .329 | .433 | .526 |
| ConE | **.496** | **.453** | **.515** | **.579** |

Table 6: Results of ablation study on rotation, for knowledge graph completion task on WN18RR. ConE w/o rotation is the model that applies restricted rotation in the whole embedding space for hierarchical relations.

(a) On knowledge graph completion

| Model | MRR | H@1 | H@3 | H@10 |
|---|---|---|---|---|
| Orthogonal | .493 | .449 | .512 | .577 |
| Overlapping | **.495** | **.451** | **.513** | **.582** |

(b) On ancestor-descendant completion, in mAP metric

| Model | 0% | 50% | 100% |
|---|---|---|---|
| Orthogonal | **.930** | **.863** | .772 |
| Overlapping | .928 | .862 | **.773** |

Table 7: Comparison between orthogonal subspaces and overlapping subspaces on WN18RR benchmark.

especially in predicting missing hierarchical relations. The results suggest that restricted rotation transformation helps model hierarchical relation patterns.

**Individual results for each hierarchical relation**. To further demonstrate that ConE can deal with multiple hierarchical relations simultaneously with our proposed restricted rotation in subspaces, we report the improvement for knowledge graph completion on each type of missing hierarchical relation after adding cone restricted rotation, shown in Table 5. We observe significant improvement on *all* hierarchical relations, which shows our way of modeling heterogeneous hierarchies to be effective. Note that up to 72% improvement is achieved for some hierarchical relation thanks to the restricted rotation operation in ConE.

### A.2.2 Ablation study on rotation transformation

To address the importance of rotation transformation in modeling hierarchical relations, we present the performance comparison between ConE that uses rotation and one that does not use rotation for hierarchical relations on WN18RR. The results in Table 6 suggest that rotation transformation for hierarchical relations is significant to the model's expressive power.

## B Strategies in assigning relation-specific subspace and embedding space curvature

We compare several strategies for assigning subspace for each hierarchical relation. For simplicity, we assign equal dimension subspaces for all hierarchical relations.

### B.1 Overlapping subspaces and orthogonal subspaces

First, we compare the results on ancestor-descendant prediction and knowledge graph completion between different subspace assigning strategies, i.e., using overlapping subspaces and using orthogonal subspaces. We conduct the experiment on WN18RR dataset. For both strategies, the embedding dimension $d = 500$ and the subspace dimension $d_s = 70$ for each hierarchical relation (7 hierarchical relations in total hence it is possible to assign orthogonal subspaces). For assigning overlapping subspaces, since it is impossible to investigate all possible combinations, we randomly choose $d_s$ out of $d$ number of hyperbolic planes to each hierarchical relation. To avoid the randomness of the results due to our method in assigning overlapping subspaces, we repeat the experiment multiple times and take the average for the final result.

**Results**. Table 7 reports the results on ancestor-descendant prediction task as well as knowledge graph completion task. Between two strategies, ConE performs slightly better on knowledge graph completion task under overlapping subspaces, while their performances are comparable on ancestor-descendant prediction task. The most significant advantage for using overlapping subspaces is that it does not suffer from limitation of subspace dimension, while for orthogonal subspaces the subspace dimension can be at most $d/n$ where $n$ is the number of hierarchical relations.

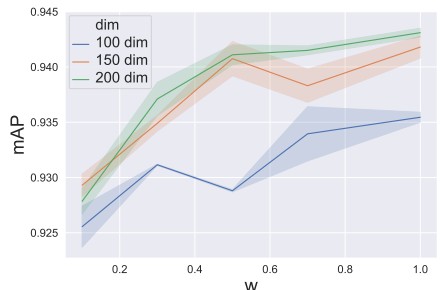
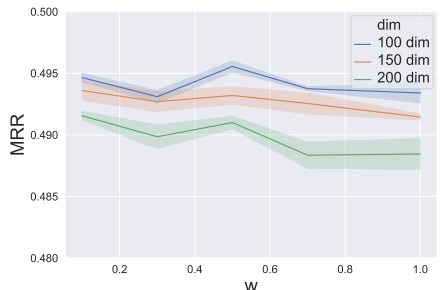

(a) Performance on ancestor-descendant prediction (0% inferred descendant pairs)

(b) Performance on knowledge graph completion task

Figure 5: Performance of two tasks on WN18RR under varying strategies, including angle loss weight $w \in \{0.1, 0.3, 0.5, 0.7, 1.0\}$, dimension of subspace $d_s \in \{100, 150, 200\}$. Due to larger number of dimensions used per subspace, we use overlapping subspace strategy to assign relation-specific subspaces.

| Model | MRR | H@1 | H@3 | H@10 |
|-------|-----|-----|-----|------|
| RotatE | .387 | .330 | .417 | .491 |
| MuRP | .465 | .420 | .484 | .544 |
| RotH | **.472** | .428 | **.490** | **.553** |
| ConE | .471 | **.436** | .486 | .537 |

Table 8: Knowledge graph completion results for low-dimensional embeddings ($d = 32$) on WN18RR. Best score in **bold** and second best underlined.

## B.2 Subspace dimension and angle loss weight

We also study the effect of subspace dimension $d_s$ and angle loss weight $w$ (in Eq. 17) on the performance of ConE. We use overlapping subspaces where we randomly choose $d_s$ out of $d = 500$ hyperbolic planes to compose the subspace for each hierarchical relation.

**Results**. Figure 5 reports the results on both tasks in curves. We notice a trade-off between two tasks for subspace dimension, where a larger dimension contributes to better performance on hierarchical task, while limiting the performance on knowledge graph completion task. With larger angle loss weight $w$, cone containment constraint is enforced more strictly, and thus the performance of ConE on hierarchical task improves as shown in Figure 5(a). On the other hand, ConE reaches peak performance on knowledge graph completion task at $w = 0.5$.

## B.3 Space curvature

Aside from setting fixed curvature $c = -1$, we also investigate on learning curvature, as [16] suggests that fixing the curvature has a negative impact on performance of RotH. With learning curvature, ConE has (MRR, H@1, H@3, H@10) = (0.485, 0.441, 0.501, 0.570), on WN18RR benchmark, lower than original ConE with fixed curvature with (MRR, H@1, H@3, H@10) = (0.496, 0.453, 0.515, 0.579). The reason why RotH [16] needs learning space curvature while ConE does not lie in the choice of embedding space: RotH uses a $2d$-dimensional hyperbolic space while ConE uses product space of $d$ hyperbolic planes. Our embedding space is less sensitive to its curvature, since for every subspace, the hierarchical structure for the corresponding single relation is less complex (than the entire hierarchy), and can thus be robust to choices of curvatures.

## C   Knowledge graph completion results in low dimensions

One of the main benefits of learning embeddings in hyperbolic space is that it can model well even in low embedding dimensionalities. We report in Table 8 the performance of ConE in the

| | Percentage of inferred descendant pairs | | | | | | | | |
|---|---|---|---|---|---|---|---|---|---|
| | WN18RR | | | DDB14 | | | GO21 | | |
| Model | 0% | 50% | 100% | 0% | 50% | 100% | 0% | 50% | 100% |
| Order | .859 | .676 | .495 | .971 | .745 | .533 | .643 | .587 | .542 |
| Poincaré | .784 | .649 | .511 | **.981** | .763 | .541 | .534 | .529 | .526 |
| HypCone | .767 | .635 | .501 | .976 | .773 | .560 | .553 | .541 | .531 |
| RotatE | .599 | .601 | .599 | .506 | .505 | .514 | .622 | .607 | .598 |
| RotH | .559 | .567 | .574 | .525 | .519 | .517 | .630 | .612 | .594 |
| ConE | **.874** | **.762** | **.657** | .979 | **.888** | **.802** | **.704** | **.653** | **.601** |
| Improvement (%) | +1.7% | +12.7% | +9.7% | -0.2% | +14.9% | +43.2% | +9.5% | +6.7% | +0.5% |

Table 9: Ancestor-descendant prediction results in AUROC. Best score in **bold** and second best underlined.

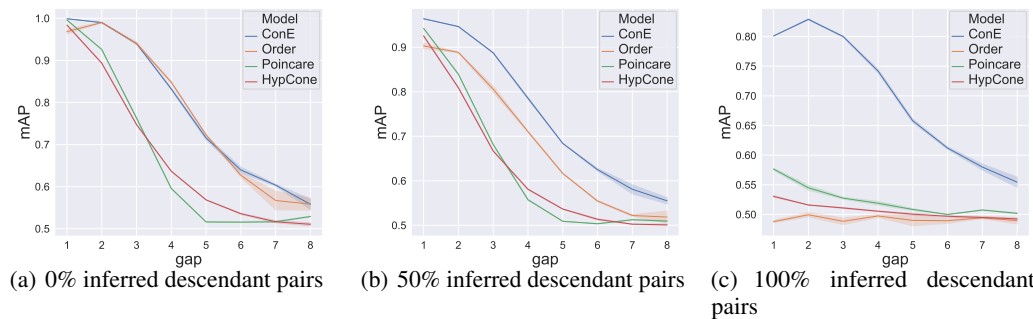

(a) 0% inferred descendant pairs    (b) 50% inferred descendant pairs    (c) 100% inferred descendant pairs

Figure 6: mAP results on ancestor-descendant prediction under different hierarchy gaps (Def. 6) on WN18RR.

low-dimensional setting for $d = 32$ on WN18RR dataset. Our performance is comparable to other hyperbolic embedding models (MuRP and RotH), while being superior to Euclidean embedding models (RotatE).

## D   Dataset details and GO21 dataset

WN18RR is a subset of WordNet [8], which features lexical relationships between word senses. More than 60% of all triples characterize hierarchical relationships. DDB14 is collected from Disease Database, which contains terminologies including diseases, drugs, and their relationships. Among all triples in DDB14, 30% include hierarchical relations.

GO21 is a biological knowledge graph containing genes, proteins, drugs and diseases as entities, created based on several widely used biological databases, including Gene Ontology [9], Disgenet [34], CTD [35], UMLS [36], DrugBank [37], ClassyFire [38], MeSH [39] and PPI [40]. It contains 80k triples, while nearly 35% of which include hierarchical relations. The dataset will be made public at publication.

## E   AUROC results and hierarchy gap studies on ancestor-descendant prediction

We show in Table 9 the results with AUROC (Area Under the Receiver Operating Characteristic curve) metric on ancestor-prediction tasks. It can be seen that the performance trend with AUROC metric is similar to that in Table 2 with mAP metric.

**Definition 6.** *Hierarchy gap. The hierarchy gap of an ancestor-descendant pair $(u, v)$ is the length of path consisting of the same hierarchical relation connecting $u$ and $v$.*

Moreover, we evaluate the classification performance of our model against other baselines over ancestor-descendant pairs with different hierarchy gaps (Def. 6), as shown in Figure 6. The trend of the curves is in line with our expectation: performance gets worse with larger hierarchy gaps.

|  | 1-Hop | | | 2-Hop | | | 3-Hop | | |
|---|---|---|---|---|---|---|---|---|---|
| Model | H@1 | H@3 | H@10 | H@1 | H@3 | H@10 | H@1 | H@3 | H@10 |
| Order [19] | 39.2% | 55.1% | 61.6% | 27.6% | 40.1% | 54.7% | 15.1% | 24.7% | 42.2% |
| Poincaré [10] | 1.5% | 3.0% | 8.0% | 31.4% | 34.6% | 38.5% | 19.5% | 23.1% | 38.5% |
| HypCone [12] | 15.0% | 30.7% | 53.0% | 16.5% | 30.1% | 43.3% | 12.4% | 38.1% | 52.0% |
| RotatE [7] | 54.7% | 63.3% | 69.7% | 20.4% | 29.0% | 35.8% | 14.6% | 18.3% | 20.7% |
| RotH [16] | 79.7% | 86.0% | 86.4% | 29.1% | 35.7% | 40.2% | 13.9% | 18.0% | 21.9% |
| ConE | **98.1%** | **99.3%** | **99.4%** | **48.6%** | **89.6%** | **97.3%** | **24.2%** | **55.6%** | **80.6%** |

Table 10: LCA prediction task results on the WN18RR dataset. N-hop means that for any pair $(u, v)$ in the test set, the true LCA $w$ has hierarchy gaps (Def. 6) at most $N$ to $u$ and $v$. The task difficulty increases as the maximum number of hops to ancestor increases. Best score in **bold** and second best underlined.

Under the setting of 0% inferred pairs, the performance of Poincaré embedding and Hyperbolic cone embedding drops dramatically as hierarchy gap increases, suggesting that transitivity is not well-preserved in these embeddings under heterogeneous setting. In all settings (0%, 50% and 100% inferred descendant pairs), ConE significantly outperforms baselines.

## F   Hierarchical analysis: LCA prediction

We further demonstrate flexibility and power of ConE using a new hierarchical task, lowest common ancestor (LCA) prediction. Given two entities, we want to find the most distinguishable feature they have in common, e.g., *LCA(WinePalm, SugarPalm)=PalmTree* in Figure 1(a). Formally, let $l_{uv}$ denote the hierarchy gap (Def. 6) between $u$ and $v$ and $l_{uv} = \infty$ if $u$ is not an ancestor of $v$, then we define $LCA(u, v) = \text{argmin}_{w \in \mathcal{E}}[(l_{wu} + l_{wv})]$. Note that if multiple common ancestors have the same sum of hierarchy gap, we consider any of them to be correct. ConE uses ranking over all entities to predict LCA, with the following scoring function for $w$ to be the LCA of $u$ and $v$:

$$\Phi_w(u, v) = \mathbf{m} \cdot (2\phi(\mathbf{w}_i) - \angle_{\mathbf{w}_i}\mathbf{u}_i - \angle_{\mathbf{w}_i}\mathbf{v}_i)_{i \in \{1, \cdots, d\}} \tag{20}$$

We evaluate the LCA prediction task on WN18RR dataset, and use the embeddings of our trained ConE model to rank and make prediction. Standard evaluation metrics including Hits at N (Hits@N) are calculated. Since no previous KG embedding method can directly perform the LCA task, we adapt them by training an MLP layer with the concatenation of the two entity embeddings as input and output the predicted entity (trained as a multi-label classification task).

**Results**. Table 10 reports the LCA prediction results. ConE can provide much more precise LCA prediction than baseline methods, and the performance gap increases as the number of hops to ancestor increases. We summarize the reasons that ConE performs superior to previous methods on LCA prediction: the task requires (1) the modeling of partial ordering for ancestor-descendant relation prediction and (2) an expressive embedding space for distinguishing the lowest ancestor. Only our ConE model is able to do both.

## G   Training details

We report the best hyperparameters of ConE on each dataset in Table 11. As suggested in [12], hyperbolic cone is hard to optimize with randomized initialization, so we utilize RotC model (which only involves rotation transformation) as pretraining for ConE model, and recover the entity embedding from the pretrained RotC model with 0.5 factor. For both the pretraining RotC model and ConE model, we use Adam [32] as the optimizer. Self-adversarial training has been proven to be effective in [7], we also use self-adversarial technique during training for ConE with self-adversarial temperature $\alpha = 0.5$.

**Knowledge graph completion**. Standard evaluation metrics including Mean Reciprocal Rank (MRR), Hits at N (H@N) are calculated in the filtered setting where all true triples are filtered out during ranking.

In our experiments, we train and evaluate our model on a single GeForce RTX 3090 GPU. We train the model for 500 epochs, 1000 epochs, 100 epochs and 600 epochs on WN18RR, DDB14, GO21

| Dataset | embedding dim | learning rate | batch size | negative samples | subspace dim | angle loss weight |
|---------|---------------|---------------|------------|------------------|--------------|-------------------|
| WN18RR | 500 | 0.001 | 1024 | 50 | 100 | 0.5 |
| DDB14 | 500 | 0.001 | 1024 | 50 | 50 | 0.7 |
| GO21 | 500 | 0.005 | 1024 | 50 | 50 | 0.1 |
| FB15k-237 | 500 | 0.0001 | 1024 | 100 | - | - |

Table 11: Best hyperparameter setting of ConE on four datasets.

and FB15k-237 respectively, and the training procedure takes 4hrs, 2hrs, 6hrs, 6hrs on these four datasets. On knowledge graph completion task, ConE model has standard deviation less than 0.001 on MRR metric across all datasets. On ancestor-descendant classification task, ConE model has standard deviation less than 0.01 on mAP metric across all datasets.

For all baselines mentioned in our work, we also perform comprehensive hyperparameter search. Specifically, for KG embedding methods (TransE [5], RotatE [7], TuckER [14], HAKE [33], MuRP [15], RotH [16]), we search for embedding dimension in $\{100, 250, 500\}$, batch size in $\{256, 512, 1024\}$, learning rate in $\{0.01, 0.001, 0.0001\}$ and negative sampling size in $\{50, 100, 250\}$. For partial order modeling methods (Order [19], Poincaré [10], HypCone [12]), we search for embedding dimension in $\{50, 100, 250, 500\}$ and learning rate in $\{0.001, 0.0001, 0.00001\}$.

# H    Krackhardt hierarchical measurement

## H.1    Krackhardt score for the whole graph

The paper [29] proposes a set of scores to measure how hierarchical a graph is. It includes four scores: *(connectedness, hierarchy, efficiency, LUBedness)*. Each score range from 0 to 1, and higher scores mean more hierarchical. When all four scores equal to 1, the digraph is a tree, normally considered as the most hierarchical structure. We make some adjustments to the computation of the metrics from the original paper to adapt them to heterogeneous graphs.

*1. Connectedness*. Connectedness measures the connectivity of a graph, where a connected digraph (each node can reach every other node in the underlying graph) will be given score 1 and the score goes down with more disconnected pairs. Formally, the degree of connectedness is

$$connectedness = \frac{c}{n(n-1)/2} \tag{21}$$

where $c$ is the number of connected pairs and $n$ is the total number of nodes.

*2. Hierarchy*. Hierarchy measures the order property of the relations in the graph. If for each pair of nodes such that one node $u$ can reach the other node $v$, $v$ cannot reach $u$, then the hierarchy score is 1. In knowledge graph this implies that if *(u, rel, v)* $\in \mathbb{T}$ then *(v, rel, u)* $\notin \mathbb{T}$. Let $T$ denote the set of ordered pairs (u, v) such that $u$ can reach $v$, and $S = \{(v, u) | (u, v) \in T, v \text{ cannot reach } u\}$, the degree of hierarchy is defined as

$$hierarchy = \frac{|S|}{|T|} \tag{22}$$

*3. Efficiency*. Another condition to make sure that a structure is a tree is that the graph contains exactly $n - 1$ edges, given $n$ number of nodes. In other word, the graph cannot have redundant edges. The degree of efficiency is defined as

$$efficiency = 1 - \alpha \cdot \frac{m - (n-1)}{(n-1)(n-2)/2} \tag{23}$$

where $m$ is the number of edges in the graph. Numerator $m - (n-1)$ is the number of redundant edges in the graph while denominator $(n-1)(n-2)/2$ is the maximum number of redundant edges possible. In the original paper [29], $\alpha$ is set to 1, in our case we take $\alpha = 500$ to make the gap larger since common knowledge graph are always sparse.

*4. LUBedness*. The last condition for a tree structure is that every pair of nodes has a least upper bound, which is the same as our defined LCA concept (in Sec. F) in knowledge graph case. Different from the homogeneous setting in [29], we still restrict LCA to a single relation (same relation on the

paths between the pair of nodes and their LCA), since heterogeneous hierarchies may exist in a single KG. Let $T = \{(u, v) | (u, v) \text{ has a } LCA\}$, then the degree of LUBedness is defined as

$$LUBedness = \frac{|T|}{n(n-1)} \tag{24}$$

## H.2 Hierarchical-ness scores for each relation

Here we introduce the Hierarchical-ness scores for each relation, which is a modified version of original Krackhardt scores on the induced subgraph of a relation. We observe, using the groundtruth hypernym, hyponym and non-hierarchical relations in existing datasets (WN18RR, DDB14, GO21), that the Hierarchical-ness scores for hypernym, hyponym and non-hierarchical relations can be easily separated via decision boundaries. To apply ConE on a dataset where the type of relation is not available, we can compute the Hierarchical-ness scores of the relations, and classify the hierarchical-ness of the relations via the decision boundaries.

Here we introduce the computation of our Hierarchical-ness scores, which contain two terms: *(asymmetry, tree_likeness)*.

*1. Asymmetry*. The *asymmetry* metric is the same as *hierarchy* metric in Krackhardt scores.

*2. Tree_likeness*. The *tree_likeness* metric is adapted from the *LUBedness* metric in Krackhardt scores where three adjustments are made:

**(a)** The subgraph induced by a single relation is not guaranteed to be connected, and forest is a typical hierarchical structure in such a disconnected graph. We cannot make sure every pair of nodes are in the same tree, and thus we evaluate on all connected pairs and check whether they have an LCA. Let $P$ denote the set of pairs $(u, v)$ such that $u$ and $v$ are connected, and the set $Q = \{(u, v) | (u, v) \in P \text{ and } (u, v) \text{ has a } LCA\}$. Then our new *LUBedness'* for disconnected graph is calculated as

$$LUBedness' = \frac{|Q|}{|P|} \tag{25}$$

**(b)** We want to distinguish true hierarchical relations from common 1-N relations, where the transitivity property may not hold (for example, *participants* of some event entity is a 1-N relation, yet it does not define a partial ordering since the head entity and tail entity are not the same type of entities). This kind of relation can be characterized by 1-depth trees in their induced subgraph, while hierarchical relations usually induce trees of greater depth. Hence we add punishment to the induced subgraphs containing mostly 1-depth trees to exclude non-hierarchical 1-N relations. In particular, let $E$ denote the set of edges, and $S = \{u | \exists v : (u, v) \in E \text{ or } (v, u) \in E\}$, $T = \{u | \exists v : (u, v) \in E \text{ and } (v, u) \in E\}$. If 1-depth trees are prevalent in the structure, then $|T|$ is approximately 0. We define the punishment decaying factor (lower means more punishment):

$$d = \frac{|T|}{|S|} \tag{26}$$

**(c)** *LUBedness* metric also depends on the direction of the relation, since LCA exists only if the relations are hyponym type (pointing from parent node to child nodes) while hypernym type relation can also define a partial ordering and considered as hierarchical relation. Hence for each relation, we define two induced graphs $G$ and $G_{rev}$, $G$ in original direction and $G_{rev}$ in reversed direction. We calculate the *LUBedness* metric of the two graphs, if the score of $G$ is much higher than the score of $G_{rev}$ then the relation is of hyponym type, and vice versa. We take the absolute value of $LUBedness'(G) - LUBedness'(G_{rev})$ as the score to measure the hierarchical-ness while its sign to check if it is of hypernym type or hyponym type.

Finally, our *tree_likeness* metric is calculated through

$$tree\_likeness = (LUBedness'(G) - LUBedness'(G_{rev})) / \max(1, (\log_{10}(d))^2) \tag{27}$$

We show that our Hierarchical-ness scores indicate the type of relation on WN18RR, DDB14 and GO21 datasets. In Figure 7(a), Figure 8(a), Figure 9(a), we visualize the two-dimensional scores for each relation in the three datasets. Groundtruth hypernym type relations are colored in orange and hyponym relations are colored in violet. We can see that hypernym type relations are clustered

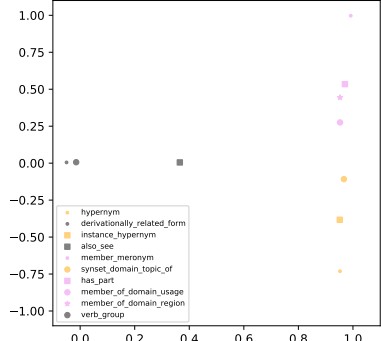

| Relation | Score | Hierarchical |
|---|---|---|
| *hypernym* | 2.0 | true |
| *member meronym* | 2.0 | true |
| *has part* | 2.0 | true |
| *member of domain region* | 1.9 | true |
| *instance hypernym* | 1.7 | true |
| *member of domain usage* | 1.5 | true |
| *synset domain topic of* | 1.2 | true |
| *also see* | 0.4 | false |
| *derivationally related form* | 0.0 | false |
| *verb group* | 0.0 | false |
| *similar to* | 0.0 | false |

(a) Hierarchical-ness score visualization for all relations.

(b) Relation type classification based on Hierarchical-ness score.

Figure 7: Results on WN18RR dataset. (a) Score *(asymmetry, tree_likeness)* as $(x, y)$ coordinate in visualization. Orange dots denote hypernym type relations, violet dots denote hyponym type relations and black dots denote non-hierarchical relations. (b) Relations above the line are predicted to be hierarchical relations, and ground-truth relation type are in the third column. All relations are correctly predicted.

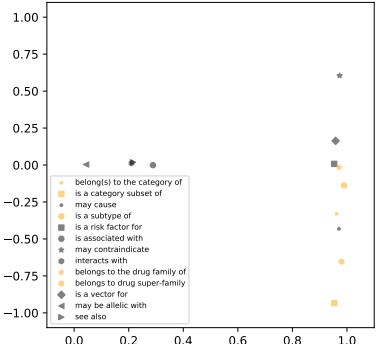

| Relation | Score | Hierarchical |
|---|---|---|
| *is a category subset of* | 2.0 | true |
| *belongs to drug super-family* | 2.0 | true |
| *has part* | 2.0 | true |
| *may contraindicate* | 1.9 | false* |
| *may cause* | 1.8 | false* |
| *belong(s) to the category of* | 1.7 | true |
| *belong(s) to the category of* | 1.2 | true |
| *is a vector for* | 1.3 | false* |
| *is a subtype of* | 1.3 | true |
| *belongs to the drug family of* | 1.1 | true |
| *is a risk factor for* | 1.0 | false |
| *see also* | 0.3 | false |
| *interacts with* | 0.3 | false |
| *may be allelic with* | 0.1 | false |
| *is an ingredient of* | 0.0 | false |

(a) Hierarchical-ness score visualization for all relations.

(b) Relation type classification based on Hierarchical-ness score.

Figure 8: Results on DDB14 dataset. (a) Score *(asymmetry, tree_likeness)* as $(x, y)$ coordinate in visualization. Orange dots denote hypernym type relations, violet dots denote hyponym type relations and black dots denote non-hierarchical relations. (b) Relations above the line are predicted to be hierarchical relations, and ground-truth relation type are in the third column. Predictions are correct except three non-hierarchical relations are inferred to be hierarchical relations, while these relations do have soft-hierarchical property.

in lower right while hyponym type relations are clustered in upper right, indicating that hyponym type relations have large *asymmetry* score and large positive *tree_likeness* score while hyponym type relations have large *asymmetry* score and large negative *tree_likeness* score. Moreover, We use *asymmetry* + |*tree_likeness*| as the total Hierarchical-ness score and set threshold 1.1 to separate hierarchical relations and non-hierarchical relations. As shown in Figure 7(b), Figure 8(b), Figure 9(b), our classification results highly conform to the groundtruth relation type.

Additionally, we use our Hierarchical-ness scores to distinguish hierarchical relations from 237 relations in FB15k-237, as shown in Figure 10(a), Figure 10(b). Since there is no labeling of relation type in FB15k-237, we do not have groundtruth. We label the relations that rank highest

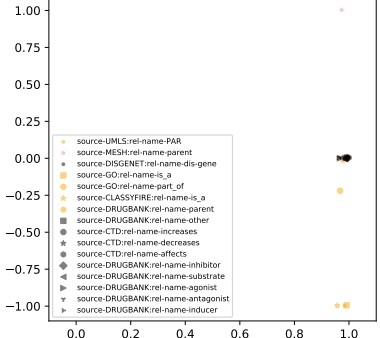

(a) Hierarchical-ness score visualization for all relations.

| Relation | Score | Hierarchical |
|---|---|---|
| *source-GO:rel-name-is_a* | 2.0 | true |
| *source-MESH:rel-name-parent* | 2.0 | true |
| *source-CLASSYFIRE:rel-name-is_a* | 2.0 | true |
| *source-UMLS:rel-name-PAR* | 2.0 | true |
| *source-GO:rel-name-part_of* | 1.2 | true |
| *rest of the relations* | < 1.1 | false |

(b) Relation type classification based on Hierarchical-ness score.

Figure 9: Results on GO21 dataset. (a) Score *(asymmetry, tree_likeness)* as $(x, y)$ coordinate in visualization. Orange dots denote hypernym type relations, violet dots denote hyponym type relations and black dots denote non-hierarchical relations. (b) Relations above the line are predicted to be hierarchical relations, and ground-truth relation type are in the third column. All relations are correctly predicted.

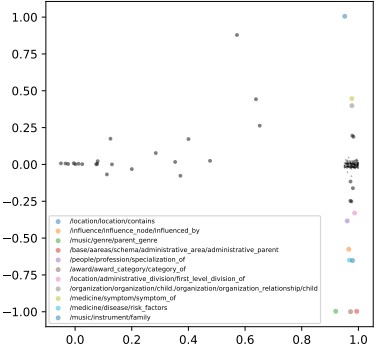

(a) Hierarchical-ness score visualization for all relations.

| Relation | Score | Hierarchical |
|---|---|---|
| *administrative_area/administrative_parent* | 2.0 | true |
| *award_category/category_of* | 2.0 | true |
| *music/genre/parent_genre* | 2.0 | true |
| *location/contains* | 2.0 | true |
| *music/instrument/family* | 1.7 | true |
| *medicine/disease/risk_factors* | 1.7 | unknown |
| *influence/influence_node/influenced_by* | 1.6 | unknown |
| *medicine/symptom/symptom_of* | 1.4 | unknown |
| *organization_relationship/child* | 1.4 | true |
| *administrative_division/first_level_division_of* | 1.3 | true |
| *rest of the relations* | < 1.1 | false |

(b) Relation type classification based on Hierarchical-ness score.

Figure 10: Results on FB15k-237 dataset. (a) Score *(asymmetry, tree_likeness)* as $(x, y)$ coordinate in visualization. Dots with non-black colors denote top hierarchical-like relations among all 237 relations (their meanings are annotated in lower left of the figure). (b) Relations above the line are predicted to be hierarchical relations, and ground-truth relation type in column 3 are manually labeled.

on Hierarchical-ness score and discover that they are indeed hierarchical relations (suggested by keywords in their name, such as "child", "parent").