# OpenReview forum: "Modeling Heterogeneous Hierarchies with Relation-specific Hyperbolic Cones"
_NeurIPS.cc/2021/Conference — NeurIPS 2021 Poster_

### Official Review · Reviewer_LaLE · 2021-07-06

**Rating:** 7
**Confidence:** 5

**Summary:**

This work proposes ConE, a knowledge graph embedding model which embeds entities into hyperbolic cones and represents relations as transformations between the cones. Non-hierarchical relations are modelled using rotations and hierarchical relations using restricted rotations, which impose the cone containment constraint. The proposed method is evaluated on the KG completion and ancestor-descendant prediction tasks.

**Limitations And Societal Impact:**

Yes.

**Main Review:**

Strengths
- The proposed model is a relatively novel approach for modelling both hierarchical and non-hierarchical relations in knowledge graphs, based on hyperbolic cones by Ganea et al., 2018.
- The ability to do ancestor-descendant prediction is an important improvement compared to standard KG completion models.
- The performance on KG completion is comparable to existing state-of-the-art models.

Weaknesses
- The proposed method requires a priori knowledge of whether a relation is hierarchical or not, and not just that, but also knowing the direction of hierarchy (i.e. head->tail or tail->head). None of the previous methods (e.g. MuRP and RotH) require this knowledge, which gives ConE unfair advantage over those.
- Despite having access to extra information, the proposed model does not outperform RotH/AttH on FB15k-237 and WN18RR, which are standard KG completion datasets.


Detailed questions/comments:
1. In the abstract, you claim that "curent KG embeddings...fail to model multiple heterogeneous hierarchies that exist in a single KG.". However, this is factually incorrect, since there has been several works for modelling hiearchical relations in knowledge graphs, such as MuRP and RotH/AttH.
2. Again, same applies to the introduction, i.e. "These methods can only be applied to graphs with a single hierarchical relation type, and are thus not suitable to real-world knowledge graphs with multiple hierarchical relations." - both MuRP and RotH/AttH have been specifically designed to model multiple hierarchies. Later in that some paragraph, you contradict yourself by mentioning MuRP and RotH.
3. Have you tried learning the space curvature? Chami et al. show that fixing the curvature has a negative impact on performance.
4. Your method assumes a priori knowledge on whether a relation is hierarchical or not, but in most real world KGs, this information isn't available. What would happen for KGs where the number of relations is really large and checking whether a relation is hierarchical or not is infeasible?
5. Do you have any intuition on why the multi-relational RotH performs much worse than the single-relational Poincaré embedding (Nickel & Kiela, 2017) on the ancestor-descendant prediction task ?

**Time Spent Reviewing:**

2

---

> ### Author Response · Authors · 2021-08-10
> **Response to review (prior knowledge requirement was already addressed in appendix in submission)**
>
> We thank the reviewer for the comprehensive and insightful comments. Many of the points help us to better clarify details of the paper, and allow us to provide more insights to the readers. We provide detailed explanations in the following.
>
> **Q1: ConE needs prior knowledge on relation types.**
>
> **A1:** We point out that as discussed in Appendix section I.2, the hierarchical-ness information is widely available and can be accurately and efficiently computed even if such information is NOT provided. Using the inferred hierarchical-ness of relations has no performance impact on datasets (e.g. WN18RR, FB15k-237). We did not include it in main text since it's orthogonal to the main method of the paper, which aims to capture transitive (partial ordering) properties of hierarchical relations.
> Our automatic detection for relation types is proved effective for KGs where the number of relations is really large and checking whether a relation is hierarchical or not is infeasible, referring to the results on FB15k-237 in section I.2.
> Also note that, existing method has no way to incorporate this information due to the inability to model hierarchical relations in an effective way. So even with such information being available, the performance of previous methods (RotH, RotE, TuckER etc.) will not change.
>
> **Q2: Performance on FB15k-237 and WN18RR.**
>
> **A2:** FB15k-237 is non-hierarchical (indicated by the Krackhardt scores) and thus not suitable for ConE’s design. We outperform existing Sota on link prediction through all three hierarchical KG datasets (although only by a small margin for WN18RR). We further emphasize that the extra information on hierarchical-ness is essential for the model to perform hierarchical reasoning tasks which cannot be done by all existing KG embedding models. Our model also achieves state-of-the-art for ancestor-descendant prediction task and LCA prediction task across a different set of baselines.
>
> **Q3: Detailed comments on writing. "In the abstract, you claim that "curent KG embeddings...fail to model multiple heterogeneous hierarchies that exist in a single KG.". However, this is factually incorrect, since there has been several works for modelling hiearchical relations in knowledge graphs, such as MuRP and RotH/AttH."**
>
> **A3:** In the abstract and introduction, when we are saying “model hierarchy”, we are emphasizing that **partial ordering property** needs to be modeled: the model needs to infer A->C through A->B and B->C  (-> refers to the parent/child hierarchical relations).
> Simply using hyperbolic space is not sufficient to **model hierarchy**.
> As illustrated in the abstract, the key property of hierarchical relations is that they induce a partial ordering, which needs to be correctly modeled for hierarchical reasoning. Current KG embedding works, such as RotH [2] and MuRP [4], cannot faithfully capture the partial ordering property because of the limitation of the rotation/translation transformations, and thus fail to model multiple hierarchies by our definition. We will clarify this confusion in the revision.
>
> **Q4: Learning of space curvature.**
>
> **A4:** Thank you for this good suggestion. We tried trainable curvature in ConE and the knowledge graph completion results on WN18RR are as follows:
>
> |  MRR  |  H@1  |  H@3  |  H@10 |
> |:-----:|:-----:|:-----:|:-----:|
> | 0.485 | 0.441 | 0.501 | 0.570 |
>
> The performance is not better than our ConE model with constant curvature $c=-1$. The reason why RotH [1] needs learning space curvature while ConE does not lie in the choice of embedding space: RotH uses a $2d$-dimensional hyperbolic space while ConE uses product space of $d$ hyperbolic planes. Our embedding space is less sensitive to its curvature, since for every subspace, the hierarchical structure for the corresponding single relation is less complex (than the entire hierarchy), and can thus be robust to choices of curvatures.
> We thank reviewer for the suggestion, and will incorporate this discussion in the revision.
>
> **Q5: More insights on the results of ancestor-descendant prediction.**
>
> **A5:** Indeed even the best hierarchical KG embedding method like RotH [1] cannot outperform the basic hierarchy modeling approach (Poincaré embedding [2]) in ancestor-descendant prediction task, because RotH only focuses on relation transformation and the rotation transformation it uses fails to preserve the partial ordering property. Poincaré embedding is able to preserve the level of hierarchy through the embedding norm by its training objective, while RotH cannot, by a keen observation on its trained embeddings. In contrast, our method is shown to effectively model partial ordering property of heterogeneous hierarchical relations and therefore combining the advantages of both Sota KG embedding approaches, as well as explicit hierarchy modeling approaches (Poincaré [2], Cone [3] etc.).
>
>
> [1] I. Chami, A. Wolf, D.-C. Juan, F. Sala, S. Ravi, and C. Ré, “Low-dimensional hyperbolic knowledge graph embeddings,” in ACL, 2020.
>
> [2] M. Nickel and D. Kiela, “Poincaré embeddings for learning hierarchical representations,” in NeurIPS, 2017.
>
> [3] O.-E. Ganea, G. Bécigneul, and T. Hofmann, “Hyperbolic entailment cones for learning hierarchical embeddings,” in ICML, 2018.
>
> [4] I. Balazevic, C. S. Allen, and T. Hospedales, “Multi-relational poincaré graph embeddings,” in NeurIPS, 2019.

---

> > ### Comment · Reviewer_LaLE · 2021-08-17
> > **Question regarding performance on FB15k-237**
> >
> > Thank you for the detailed response.
> >
> > In A2, you claim that "FB15k-237 is non-hierarchical (indicated by the Krackhardt scores) and thus not suitable for ConE’s design.". However,  shouldn't ConE's design be capable of capturing both by modelling non-hierarchical relations using rotations and hierarchical relations using restricted rotations and should thus be superior to RotH/AttH which can only model hierachy?

---

> > > ### Author Response · Authors · 2021-08-23
> > > **Performance on hierarchical and non-hierarchical relations in FB15k-237**
> > >
> > > RotH uses rotation transformation in a $2d$-hyperbolic space to model all relations, and thus is not able to model the partial ordering property. It is superior to Euclidean embedding models (RotatE, etc) in hierarchical graphs due to its hyperbolic embedding space, which is more suitable to preserve hierarchical structure of the graph. For non-hierarchical relations, ConE uses rotation which is close to the characterization in RotH. The only difference lies in the operating space: our model operates in the product space of $d$ hyperbolic planes, which is critical in modeling partial ordering property for heterogeneous hierarchies. We do not claim that ConE is superior to RotH in modeling non-hierarchical relations, but rather we emphasize that restricted rotation transformation, one of our main contributions, is superior to RotH in modeling hierarchical relations, since it can successfully model the partial ordering property. We support this by giving the performance comparison on hierarchical relations and non-hierarchical relations in FB15k-237:
> > >
> > > | Hierarchical |  MRR  |  H@1  |  H@3  |  H@10 |
> > > |:-----:|:-----:|:-----:|:-----:|:-----:|
> > > | RotH | 0.353 | 0.251 | 0.386 | 0.566 |
> > > | ConE | **0.361** | **0.256** | **0.397** | **0.585** |
> > > | Improvement | +2.3% | +2.0% | +2.8% | +3.4% |
> > >
> > > | Non-hierarchical |  MRR  |  H@1  |  H@3  |  H@10 |
> > > |:-----:|:-----:|:-----:|:-----:|:-----:|
> > > | RotH | 0.341 | 0.246 | 0.376 | 0.534 |
> > > | ConE | **0.344** | **0.248** | **0.380** | **0.539** |
> > > | Improvement | +0.9% | +0.8% | +1.1% | +0.9% |
> > >
> > > Our superior in modeling hierarchical relations does not result in outperforming RotH on FB15k-237 by a large margin because the ratio of hierarchical relations among all triples in FB15k-237 is only ~2.5%.

---

> > > > ### Comment · Reviewer_LaLE · 2021-08-31
> > > > **Thanks for the clarification**
> > > >
> > > > Thank you for your response and the additional experiment. Your answer addressed my concern, so I have decided to raise my score to 7.

---

### Official Review · Reviewer_G45q · 2021-07-16

**Rating:** 5
**Confidence:** 4

**Summary:**

The authors propose ConE (Cone Embedding) to simultaneously model multiple hierarchical and non-hierarchical relations in a knowledge graph. ConE represents entities as hyperbolic cones and models relations as transformations between the cones. Experiments demonstrate that ConE outperforms many baselines on standard knowledge graph benchmarks.

**Limitations And Societal Impact:**

Yes. The authors addressed the limitations and potential negative societal impact of their work.

**Main Review:**

Originality:
The idea of modeling multiple hierarchical and non-hierarchical relations simultaneously in a knowledge graph is interesting.

Quality:
My concerns are as follows.
1. The authors may want to explain why ConE performs on par with RotH [1] on WN18RR and FB15k-237 (see Table 1).
2. Some related works are missing. The authors may want to compare ConE with more works related to hierarchy modeling [2,3].
3. The distance scoring function in Equation (14) may not make sense. The authors claim that the rotation transformation $f_1$ cannot explictly model hierarchical relations, while the authors compute $f_1$ in Equation (14) for the hierarchical relations in some cases, e.g. $m_1=1$ and $m_i=0,i=2,3,\dots,d$.

Clarity:
The authors may want to clarify the implementation of ConE. The restricted rotation transformation in Equation (12) and the angle loss in Equation (16) are only defined for the hyponym relation. However, the information about hyponym or hypernym relation is usually unavailable in knowledge graphs, e.g. FB15k-237. The authors may want to illustrate how to implement the restricted rotation transformation and the angle loss in practice.

Significance:
The work aims to model hierarchical structures in knowledge graphs, which is an important topic.

[1] I. Chami, A. Wolf, D.-C. Juan, F. Sala, S. Ravi, and C. Ré, “Low-dimensional hyperbolic knowledge graph embeddings,” ACL, 2020.

[2] Z. Zhang, J. Cai, Y. Zhang, and J. Wang, “Learning hierarchy-aware knowledge graph embeddings for link prediction,” in Proc. AAAI, 2020, pp. 3065–3072

[3] Z. Zhang, F. Zhuang, M. Qu, F. Lin, and Q. He, “Knowledge graph embedding with hierarchical relation structure,” in Proc. Conf. Empirical Methods Natural Lang. Process., 2018, pp. 3198–3207.


**Time Spent Reviewing:**

11 hours

---

> ### Author Response · Authors · 2021-08-10
> **Response regarding baseline comparison and distance scoring function**
>
> We thank the reviewer for the careful reading and thorough comments. According to the reviewer’s valuable suggestions, we conduct more experiments and clarify the reviewer’s concerns below.
>
> **Q1: Performance on knowledge graph completion.**
>
> **A1:** As the Krackhardt scores indicate, FB15k-237 tends to be non-hierarchical. ConE is carefully designed to model the hierarchies on hierarchical datasets, which explains why it does not perform superior to other baselines on FB15k-237.
> For WN18RR, our method is only slightly better than RotH, which is already the best-performing method on hierarchical KGs. However, none of the methods, including RotH, can simultaneously perform hierarchical reasoning taks, including ancestor-descendant prediction and LCA prediction. We highlight that we also achieve state-of-the-art in ancestor descendant prediction and LCA prediction, compared to all existing methods, such as Poincaré embedding and hyperbolic cone embedding. We note that our performance is consistently better than all baselines on all hierarchical KGs.
>
> **Q2: More KG embedding baselines.**
>
>
> **A2:** Thanks for your suggestion. Here we present the knowledge graph completion result of HAKE [1] (best out of dimension in {100, 250, 500}).
>
> | WN18RR |  MRR  |  H@1  |  H@3  |  H@10 |
> |--------|:-----:|:-----:|:-----:|:-----:|
> | HAKE   | 0.496 | 0.451 | 0.513 | **0.582** |
> | ConE   | 0.496 | **0.453** | **0.515** | 0.579 |
>
> | DDB14 |  MRR  |  H@1  |  H@3  |  H@10 |
> |-------|:-----:|:-----:|:-----:|:-----:|
> | HAKE  | 0.217 | 0.146 | 0.237 | 0.361 |
> | ConE  | **0.231** | **0.161** | **0.252** | **0.364** |
>
> | GO21 |  MRR  |  H@1  |  H@3  |  H@10 |
> |------|:-----:|:-----:|:-----:|:-----:|
> | HAKE | 0.169 | 0.104 | 0.185 | 0.295 |
> | ConE | **0.211** | **0.140** | **0.237** | **0.347** |
>
> | FB15k-237 |  MRR  |  H@1  |  H@3  |  H@10 |
> |-----------|:-----:|:-----:|:-----:|:-----:|
> | HAKE      | 0.341 | 0.243 | 0.378 | 0.535 |
> | ConE      | **0.345** | **0.247** | **0.381** | **0.540** |
>
> Our model performs consistently better than HAKE across all datasets. [2] is an early work on KG embedding and its performance is clearly below the more recent methods which we are comparing.
> Similar to RotH, [1] and [2] present KG embedding models with more suitable transformations to characterize hierarchical relations, yet they still cannot model the partial ordering property of hierarchical relations, and are thus not capable of performing hierarchical reasoning tasks. We will cite these works and add their results in the revision.
>
> **Q3: Distance scoring function in Equation (14).**
>
> **A3:** This is a good question. We emphasize that our design makes sense and is crucial in capturing not just the transitivity, but also other aspects of hierarchical relations. The rotation transformation $f_1$ is also of great significance in modeling hierarchical relations. Our design of hierarchical transformations is effective due to its ability to capture both the transitivity aspect of hierarchical relations, as well as other properties of the relation.
> Intuitively, restricted rotation $f_2$ is designed to capture the hierarchical pattern of the relation by preserving partial ordering, while rotation $f_1$ can model other relation patterns such as symmetric pattern and composition pattern due to its expressiveness (see proof in Appendix A).
>
> **Q4: The implementation of ConE while relation types are not known.**
>
> **A4:** We introduce the Hierarchical-ness scores in Appendix I.2, which is proven effective in distinguishing hypernym / hyponym relations, as suggested by the results across four datasets. In practice, we can use such scores to first give us the information on relation types (the calculation itself is much efficient) and then implement different transformations to different relations according to ConE model.
>
> [1] Z. Zhang, J. Cai, Y. Zhang, and J. Wang, “Learning hierarchy-aware knowledge graph embeddings for link prediction,” in Proc. AAAI, 2020, pp. 3065–3072
> [2] Z. Zhang, F. Zhuang, M. Qu, F. Lin, and Q. He, “Knowledge graph embedding with hierarchical relation structure,” in Proc. Conf. Empirical Methods Natural Lang. Process., 2018, pp. 3198–3207.

---

> > ### Comment · Reviewer_G45q · 2021-08-18
> > **Thanks for the authors' rebuttal**
> >
> > I have read the authors' response and all the other reviewers' comments. However, my major concerns have not been properly addressed.
> >
> > 1. The marginal performance improvements on FB15k-237 do not convince me of ConE's advantages, i.e., simultaneously modeling multiple hierarchical and non-hierarchical relations. Figures 9(a) and 9(b) in Appendix show that there exist many hierarchical and non-hierarchical relations in FB15k-237. We expect that ConE outperforms RotH---which is designed only for hierarchical relations---by a large margin while it does not.
> > 2. The authors point out that "the rotation transformation $f_1$ is also of great significance in modeling hierarchical relations" in the rebuttal, which is inconsistent with the claim that "$f_1$ cannot be directly applied to model hierarchical relations, because rotation does not obey transitive property" in Line 177 in the paper. Moreover, the authors may want to provide theoretical or empirical results to support the claim that "the rotation transformation $f_1$ is also of great significance in modeling hierarchical relations".

---

> > > ### Author Response · Authors · 2021-08-23
> > > **Performance on FB15k-237 and significance of rotation transformation**
> > >
> > > **Q1: Performance on FB15k-237.**
> > >
> > > **A1:** Both ConE and RotH use rotation transformation to model non-hierarchical relations, due to its expressiveness and the ability to characterize common relational patterns. Therefore it is not precise to claim that RotH is designed only for hierarchical relations. Using hyperbolic space only indicates that RotH can capture tree-like structure better, but rotation operation itself is more suitable for non-hierarchical relations. This is also the reason why RotH cannot perform ancestor-descendant prediction.
> > >
> > > As mentioned in rebuttal, our contribution is to use cone containment in subspaces to capture the partial ordering property of heterogeneous hierarchical relations, and propose the restricted rotation transformation that naturally preserve the partial ordering of cones. Empirically, we find that by characterizing partial ordering of hierarchical relations in the embedding space, the model has superior performance on knowledge graph completion, especially on inferring missing hierarchical relations (see ablation study in Appendix D, Table 6). Similarly, we find that on FB15k-237, ConE performs superior to RotH on hierarchical relations:
> > >
> > > | Hierarchical |  MRR  |  H@1  |  H@3  |  H@10 |
> > > |:-----:|:-----:|:-----:|:-----:|:-----:|
> > > | RotH | 0.353 | 0.251 | 0.386 | 0.566 |
> > > | ConE | **0.361** | **0.256** | **0.397** | **0.585** |
> > > | Improvement | +2.3% | +2.0% | +2.8% | +3.4% |
> > >
> > > | Non-hierarchical |  MRR  |  H@1  |  H@3  |  H@10 |
> > > |:-----:|:-----:|:-----:|:-----:|:-----:|
> > > | RotH | 0.341 | 0.246 | 0.376 | 0.534 |
> > > | ConE | **0.344** | **0.248** | **0.380** | **0.539** |
> > > | Improvement | +0.9% | +0.8% | +1.1% | +0.9% |
> > >
> > > Since hierarchical relations only take up ~2.5% of all triples, the performance gain on hierarchical relations is not obvious.
> > >
> > > **Q2: Significance of rotation transformation.**
> > >
> > > **A2:** The reviewer's misunderstanding comes from the subtle point that $f_1$ and $f_2$ are for 2 different aspects of hierarchical relations.  The use of $f_2$ is to preserve partial ordering of a hierarchical relation in its **relation-specific subspace**. But this alone is not enough: there are other properties of the same relation such as composition, symmetry etc. that cannot be modeled by $f_2$. Here is how $f_1$ comes into play for capturing these properties aside from partial ordering, in the complement space. When we say both are of great significance (and hence both appear in eq 14), we mean that both partial ordering and other aspects (e.g. compositionality) are crucial. We cannot model with $f_1$ nor $f_2$ alone. This is not contradicting to the claim that $f_1$ cannot model partial ordering: partial ordering is already taken care of by $f_2$, while $f_1$ focuses on other aspects of the relation explained above. Note that on ancestor-descendant prediction, ConE examines cone containment only in the subspace that $f_2$ operates in, as indicated by the scoring function Equation (16), and thus will not be influenced by the rotation transformation $f_1$ in the complement space.
> > >
> > > We explain why we use restricted rotation in subspaces instead of in the entire embedding space in the first paragraph of section 3.3. Different subspaces allow ConE to model heterogeneous hierarchies preserved in each subspace, and rotation transformation in the complement space improves the expressive power of the model. Here we present the performance comparison between ConE that uses rotation and one that does not use rotation for hierarchical relations on WN18RR.
> > >
> > > | WN18RR |  MRR  |  H@1  |  H@3  |  H@10 |
> > > |:-----:|:-----:|:-----:|:-----:|:-----:|
> > > | ConE w/o rotation | 0.397 | 0.329 | 0.433 | 0.526 |
> > > | ConE w/ rotation | **0.496** | **0.453** | **0.515** | **0.579** |
> > >
> > > The result suggests that rotation transformation for hierarchical relations is significant to the model’s expressive power. We will add this ablation study in the revision.

---

### Official Review · Reviewer_odL8 · 2021-07-16

**Rating:** 7
**Confidence:** 4

**Summary:**

This paper proposes a knowledge graph (KG) embedding method that can simultaneously model hierarchical (e.g., is-a) and non-hierarchical (e.g., friend-of) relations. The authors propose to embed entities into hyperbolic space. Inspired by Hyperbolic Entailment Cones (Ganea et al., 2018), they apply the cone containment constraint to relation-specific subspaces to deal with hierarchical relations. Inspired by RotatE (Sun et al., 2019) and RotH (Chami et al., 2020), they use hyperbolic cone rotations from the head entity to the tail entity to model non-hierarchical relations.

Experiments are conducted on two tasks: KG completion and hierarchical reasoning. Results show that the proposed method can outperform both Euclidean and hyperbolic KG embedding baselines. Results in low dimensions, ablation studies, and results on a biological dataset are further shown in the supplementary material.

**Limitations And Societal Impact:**

Please see the main review.

**Main Review:**

The paper is well-written with solid and comprehensive evaluations, reflecting authors' massive effort.

Although some ideas are directly borrowed from existing studies (e.g., Hyperbolic Entailment Cones and RotH), the general motivation of simultaneously modeling hierarchical and non-hierarchical relations is intuitive. Also, the idea of modeling heterogeneous hierarchical relations is well-motivated. Although hypernymy heterogeneity has been studied in taxonomy-related studies [1], using relation-specific subspaces to model it in the hyperbolic space is novel from my perspective.

Some questions and concerns:
- Statistical significance tests are missing in the experiments. For example, in Table 2, the gap between your methods and the best baseline is quite subtle when there are 0% inferred descendant pairs.

- Although the quantitative evaluation is comprehensive, it would be better to see some case studies and visualization results. For example, does your embedding result really preserve the cone containment constraint? For symmetric relations, does the rotation preserve the symmetry?

[1] Expanding Taxonomies with Implicit Edge Semantics. WWW 2020.

**Time Spent Reviewing:**

3

---

> ### Author Response · Authors · 2021-08-10
> **Response to suggested improvements**
>
> We thank the reviewer for the positive feedback, and the appreciation of our effort in demonstrating the approach through analysis and experiments. We are glad that the reviewer found it to be well-written, solid and comprehensive. We address the reviewer’s suggestions below.
>
> **Q1: Statistical significance tests in experiments.**
>
> **A1:** Currently, we address the standard deviation in Appendix H: On knowledge graph completion task, ConE model has standard deviation less than 0.001 on MRR metric across all datasets. On the ancestor-descendant prediction task, ConE model has standard deviation less than 0.01 on mAP metric across all datasets. Thanks for the suggestion,  and we will address it in a more conspicuous place in revision.
>
> **Q2: Case study and visualization results of trained embedding.**
>
> **A2:** Indeed, we have visualized the embedding in a particular hyperbolic plane for some hierarchical relation (although we didn't put it in paper due to space). As expected, the embeddings of high-level entities are close to the center of the hyperbolic plane, while embeddings of their descendant entities fall in their hyperbolic cones. In comparison, RotH visualization of the entities exhibit less hierarchical structure, since it only uses rotation to model relations (which does not preserve transitive closure, in contrast to our restricted rotation of cones).
> We recognize its importance to better understanding of the method, and will include the visualizations and analyses in the final version.

---

### Official Review · Reviewer_s4ku · 2021-07-19

**Rating:** 7
**Confidence:** 3

**Summary:**

Methods for embedding hypergraph-based knowledge bases, where different edge types encode different relation types, have a long history in machine learning. Alternately, there has been a more recent interest in geometric embedding approaches for knowledge bases, which associate more complex geometric objects, such as hyperbolic cones, with knowledge base entities, due to the favorable geometric properties of hyperbolic space for modeling hierarchies. However, the latter approaches have usually been limited to representing only single relation types, due to representing the relation of interest with some geometric property of the embedding space such as enclosure between entity cones. This work attempts to bridge the gap between the two approaches by modeling relations as transformations between hyperbolic cones, specifically containment in different subspaces.

**Limitations And Societal Impact:**

Experiments seem adequate to address limitations, aside from the questions about interactions between transitive and non-transitive parts raised above.

**Main Review:**

UPDATE: The authors have addressed my concerns about interactions between the transitive and non-transitive parts of the model and I am raising my score accordingly.


**Originality/Quality**

As the paper points out, there have been a couple efforts previously to produce multi-relational hyperbolic cone embeddings. However, they argue that approaches based on unrestricted Möbius matrix-vector product and addition (MuRP) cannot capture transitivity. In response, they introduce a “restricted rotation” operation that is specified when relations are known to be transitive, which scales the rotated cone to still lie within the parent cone. Transitive relations are modeled as restricted rotation in one subset of dimensions, while non-transitive ones are modeled as general rotations in another subset.

It is unclear how this differs from training two different models and sets of embeddings, one which corresponds to the transitive relations and another which corresponds to the non-transitive relations. Is this correct? This makes the claim of using a “single” model a bit vacuous if these different relation types never interact. The results in experiments are good and the experiments are extensive, but the technical contribution seems fairly minimal.


**Clarity**

The paper is well written but could be better organized, it is hard to follow exactly how the proposed model differs from previous work, and the focus on one relation at a time in the exposition makes it often hard to initially understand how the proposed model differs from simply training a separate model for each relation, until the “subspace allocation” section of 3.3.

**Significance**

Both knowledge graph embedding and nonstandard embedding geometries are popular areas of inquiry, and doing principled multi relational modeling in hyperbolic space is also interesting. The use of the restricted rotation is a good tool for practitioners to have in their arsenal, but the synthesis between the transitive and non-transitive relations is lacking.


**Time Spent Reviewing:**

1

---

> ### Author Response · Authors · 2021-08-10
> **Response to the concern: difference to training 2 separate models and interaction between hierarchical / non-hierarchical relations.**
>
> We thank the reviewer for the constructive comments.
> The major concern of the reviewer is that if our proposed approach is different from using 2 different models and sets of embeddings for hierarchical and non-hierarchical relations.
>
> We note that there is a misunderstanding, and that our approach is very different: in our model, both kinds of relations are simultaneously learned in the same embedding space, with different subspaces corresponding to different hierarchical relations..
> Since all entities are embedded in the same embedding space, hierarchical and non-hierarchical relations could “interact” in the same embedding space through their common entities. E.g. entity A could be related to entity B with a hierarchical relation; and entity B is related to entity C with a non-hierarchical relation. The reviewer’s alternative of using 2 models will not be able to capture such interaction. We provide detailed explanations addressing the questions proposed by the reviewer below.
>
> **Q1: No explicit characterization of interactions between different relation types.**
>
> **A1:** Our model falls within the standard KG embedding framework, which do not explicitly characterize the interactions between relations, but rather characterize interactions through KG triple sampling. Specifically, the training procedure requires sampling mini-batches of independent KG triples. We note that this is common to all KG reasoning (not multi-hop) approaches.
>
> Our contribution is to design a new embedding model in the hyperbolic space that can perform link prediction and hierarchy modeling (partial-ordering of hierarchical relations) at the same time. The reviewer’s alternative approach of using 2 different models cannot work in predicting relation between two entities that are connected by paths containing hierarchical and non-hierarchical relations.
>
> Moreover, on the ancestor-descendant prediction task, we observe that our improvement is more significant as the fraction of inferred descendant pairs increases (Table 2 in paper). This shows that ConE can infer missing hierarchical links by modeling other non-hierarchical relations at the same time, which is only possible when it captures implicit interactions between different types of relations.
>
> Through close observation of the datasets, we find that such interactions between different relation types are ubiquitous. For example, in WordNet dataset, the transitive relation subClass(A, B) and non-transitive relation sisterTerm(B, C) imply subClass(A, C), such interactions can also be modeled by ConE (see detailed proof in appendix A).
>
> Finally we thank the reviewer for the feedback on clarify, and will have a brief description of subspace allocation in method overview to make it smoother to read.

---

> > ### Comment · Reviewer_s4ku · 2021-08-10
> > **clarification**
> >
> > Thank you for your response. I still am unsure about this point:
> >
> > > "We note that there is a misunderstanding, and that our approach is very different: in our model, both kinds of relations are simultaneously learned in the same embedding space, with different subspaces corresponding to different hierarchical relations.. Since all entities are embedded in the same embedding space, hierarchical and non-hierarchical relations could “interact” in the same embedding space through their common entities. E.g. entity A could be related to entity B with a hierarchical relation; and entity B is related to entity C with a non-hierarchical relation."
> >
> > This is my question --- aren't A and B interacting with each other through some subspace X, and B and C are interacting through some subspace Y which is disjoint from X? So how is this interaction being propagated from A to C? E.g., proj_X(A) should look the same even if A has been updated by some gradient that comes from C, assuming we're using linear projections.

---

> > > ### Author Response · Authors · 2021-08-11
> > > **Clarification: interaction in ConE model**
> > >
> > > Our choice of transformation is in Equation (14), where $f_1$ corresponds to restricted rotation in subspace and $f_2$ corresponds to rotation in the complemented subspace. **In ConE model, A and B interact with each other through restricted rotation in subspace X, and interact through normal rotation in the complement space**. The rotation part allows it to interact with the relation between B and C. Specifically, let S denote the embedding space, then ConE enforces rotation transformation in $S / X$ from A to B, and rotation transformation in $S / Y$ from B to C. Interactions like composition between A and C can be modeled by the composition of rotation in the subspace $(S / X) \cap (S / Y)$. Moreover, as discussed in appendix B.1, ConE uses overlapping subspace, which allows different hierarchical relations to interact not only by rotation transformation in the complement space, but by restricted rotation in the intersection of their subspaces ($X \cap Y$).

---

> > > > ### Comment · Reviewer_s4ku · 2021-08-11
> > > > **thanks!**
> > > >
> > > > Thank you for clarifying this. I did see that ConE uses overlapping subspaces for hierarchical relations, but did not see that the hierarchical relations also cause a rotation in the complement space. I will raise my score accordingly.

---

### Decision · Program_Chairs · 2021-09-27

**Decision:**

Accept (Poster)

**Comment:**

This paper proposes a knowledge graph embedding method called ConE, which can simultaneously model both hierarchical and non-hierarchical relations by embedding entities into hyperbolic cones with relations as transformations between cones. The reviewers agree that this is a strong paper, and the authors did an excellent job in their rebuttal in addressing any remaining concerns.